# The mucosal adjuvant cyclic di-GMP enhances antigen uptake and selectively activates pinocytosis-efficient cells in vivo

Steven M Blaauboer[†], Samira Mansouri[†], Heidi R Tucker, Hatti L Wang, Vincent D Gabrielle, Lei Jin*

Center for Immunology and Microbial Disease, Albany Medical College, Albany, United States

**Abstract** Effective mucosal adjuvants enhance the magnitude and quality of the vaccine response. Cyclic di-GMP (CDG) is a promising mucosal vaccine adjuvant. However, its in vivo mechanisms are unclear. Here, we showed, in mice, that CDG elicits stronger Ab and $T_H$ responses than the mammalian 2′3′-cyclic GMP-AMP (cGAMP), and generated better protection against *Streptococcus pneumoniae* infection than 2′3′-cGAMP adjuvanted vaccine. We identified two in vivo mechanisms of CDG. First, intranasally administered CDG greatly enhances Ag uptake, including pinocytosis and receptor-mediated endocytosis in vivo. The enhancement depends on MPYS (STING, MITA) expression in CD11C[+] cells. Second, we found that CDG selectively activated pinocytosis-efficient-DCs, leading to $T_H$ polarizing cytokines IL-12p70, IFNγ, IL-5, IL-13, IL-23, and IL-6 production in vivo. Notably, CDG induces IFNλ, but not IFNβ, in vivo. Our study revealed previously unrecognized in vivo functions of MPYS and advanced our understanding of CDG as a mucosal vaccine adjuvant.

*For correspondence: JINL@
MAIL.amc.edu

[†]These authors contributed
equally to this work

Competing interests: The
authors declare that no
competing interests exist.

Reviewing editor: Fiona M
Powrie, Oxford University, United
Kingdom

## Introduction

Most pathogens enter the body via mucosal surfaces. Immunization by mucosal routes is more effective at inducing protective immunity against mucosal pathogens than systemic immunization. Moreover, mucosal vaccines have the benefits of low cost and ease of administration, which make mucosal vaccines particularly suitable for developing countries and during emergency. Currently, only a dozen mucosal vaccines are approved for human use. This is largely due to problems with developing safe and effective mucosal adjuvants.

Cyclic di-GMP (CDG) is a promising mucosal vaccine adjuvant candidate (*Ogunniyi et al., 2008*; *Hu et al., 2009*; *Chen et al., 2010*; *Madhun et al., 2011*; *Gray et al., 2012*). It is ubiquitously found in bacteria, but is absent in higher eukaryotes. Yan et al. found that intranasal administration of CDG, along with the pneumococcal Ag PsaA, elicits a comparable Ag-specific Ab response, and reduces bacterial colonization to those mice immunized with cholera toxin and PsaA (*Yan et al., 2009*). Cholera toxin is the most potent experimental mucosal adjuvant. CDG also exhibits balanced $T_H$1, $T_H$2, and $T_H$17 immune responses (*Ebensen et al., 2007*, *2011*; *Gray et al., 2012*). A recent study found that CDG is a more potent activator of both $T_H$1 and $T_H$2 immune responses than LPS, CpG oligonucleotides (ODN), and aluminum salt based adjuvant in mice (*Gray et al., 2012*). Thus, CDG is an excellent mucosal vaccine adjuvant candidate promoting both strong humoral and cellular immune responses.

The mechanism by which CDG acts as a mucosal adjuvant in vivo is not known (*Chen et al., 2010*). We previously showed that MPYS-deficient mice (*Tmem173⁻/⁻*) completely lost CDG induced Ag-specific Ab and $T_H$ responses (*Blaauboer et al., 2014*). MPYS, also known as STING, MITA, and TMEM173, is a type I IFN stimulator (*Ishikawa and Barber, 2008*; *Jin et al., 2008*; *Zhong et al.,*

**eLife digest** The presence of a bacterium, virus, or other pathogen in the body generally triggers a response by the body's immune system. As well as trying to destroy the infectious agent, the immune system will also generate 'memory cells' that are primed and ready to recognize and help eliminate the pathogen if it is ever re-encountered. The parts of the invader that the memory cells recognize are called antigens.

A vaccine is a biological preparation that improves immunity to a particular disease. Vaccines normally contain a dead or weakened version of a pathogen, or its toxins or surface proteins. This exposes the immune system to the antigens in a harmless way, and creates memory cells that are able to fight off the harmful pathogen in the future before the individual becomes ill from the infection. Substances called adjuvants must also be added to many modern vaccines. Adjuvants help to present antigens to immune cells, and by doing so stimulate a stronger and more targeted immune response.

While many vaccines are currently injected, there is growing interest in developing and improving vaccines that can be inhaled. This delivers the vaccine directly to the mucosal surfaces that line the nose and lungs, which is a more effective way to produce immunity against certain bacteria and viruses. As these mucosal vaccines are also relatively cheap and easy to apply, they would also be suitable for use in developing countries and during emergencies.

Current licensed pneumococcal vaccines do not provide strong mucosal protection against the infection. As a result, pneumococcal diseases kill more people than all vaccine-preventable diseases combined. Developing safe and effective mucosal vaccine adjuvants is key to reducing the impact of pneumococcal diseases.

Cyclic di-GMP, a molecule found primarily in bacteria, is a powerful mucosal adjuvant. However, before it can be widely used in vaccines, it first needs to be known how cyclic di-GMP stimulates the immune system.

Blaauboer, Mansouri et al. studied the immune response of mice to cyclic di-GMP applied through the nose. This revealed two ways that cyclic di-GMP enhances the body's immune response to a vaccine. First, cyclic di-GMP improves the uptake of antigens by certain cells exposed to the vaccine, a process that ensures a large number of cells will alert the immune system to the perceived threat. Second, Blaauboer, Mansouri et al. explain that cyclic di-GMP selectively activates immune cells known as dendritic cells, which then produce proteins called cytokines that signal to other cells and coordinate the immune response. A gene called STING (stimulator of interferon genes) controls both cyclic di-GMP induced antigen uptake and the activation of dendritic cells. Further research into these processes is now needed to investigate whether cyclic di-GMP is a suitable mucosal pneumococcal vaccine adjuvant for humans.

2008). However, we found that type I IFN signaling is not required for the mucosal adjuvant activity of CDG in vivo (*Blaauboer et al., 2014*). CDG activates both type I IFN and NF-κB signaling (*McWhirter et al., 2009*). While MPYS is required for both CDG induced type I IFN and NF-κB activations (*Jin et al., 2011a*; *Sauer et al., 2011*), we found that these two pathways can be uncoupled in dendritic cells (DCs) and macrophages (*Blaauboer et al., 2014*). Of note, it is still unknown which cell type responds to mucosal adjuvant CDG in vivo.

In this study, we investigated how CDG promotes its mucosal adjuvant activity in vivo. We found that CDG enhances Ag uptake in vivo, and selectively activates pinocytosis-efficient DCs in vivo. Furthermore, we demonstrated that these CDG activities depend on the expression of MPYS in DCs in vivo.

## Results

### CDG is a better mucosal pneumococcal vaccine adjuvant than the mammalian cyclic dinucleotide 2′3′-cyclic GMP-AMP in mice

CDG is a potent mucosal vaccine adjuvant with activity similar to that of cholera toxin, the gold standard of a mucosal vaccine adjuvant (*Yan et al., 2009*). The 2′3′-cyclic GMP-AMP (cGAMP) is

a newly discovered mammalian cyclic dinucleotide that also has mucosal adjuvant activity in vivo (*Skrnjug et al., 2014*). Both CDG and 2′3′-cGAMP can bind MPYS in vitro (*Burdette et al., 2011*; *Gao et al., 2013a*; *Sun et al., 2013*). The 2′3′-cGAMP has a much better binding affinity to MPYS than CDG (*Gao et al., 2013c*). Furthermore, 2′3′-cGAMP induces stronger type I IFN production than CDG does in mammalian cells (*Gao et al., 2013c*). We, thus, asked if the 2′3′-cGAMP exhibits superior mucosal adjuvant activity to CDG in vivo.

We intranasally administered BALB/C mice with CDG plus OVA Ag, or 2′3′-cGAMP, plus OVA Ag three times at 2 weeks interval. The serum anti-OVA IgG1, IgG2A, and nasal IgA were quantified. Surprisingly, CDG adjuvanted vaccine induced higher Ag-specific IgG1 and IgA production than the 2′3′-cGAMP adjuvanted vaccine (*Figure 1A*). The production of OVA-specific IgG2A was similar in both vaccines (*Figure 1A*).

As a mucosal adjuvant, CDG generates balanced $T_H1$, $T_H2$, and $T_H17$ responses. We next performed the ex vivo recall assay in splenocytes from immunized mice, and examined the $T_H$ cytokine production. Again, CDG adjuvanted vaccine generated better IL-13, a $T_H2$ cytokine, and IL-17 production than the 2′3′-cGAMP adjuvanted vaccine (*Figure 1B*). The $T_H1$ cytokine, IFNγ, was similarly produced by both cyclic dinucleotides (*Figure 1B*).

We then replaced OVA Ag with pneumococcal surface protein A (PspA), a protein Ag extensively tested in various pneumococcal vaccines (*Feldman and Anderson, 2014*). We also used a different mouse strain, C57BL/6, to repeat the immunization experiment. We found that CDG adjuvanted PspA based pneumococcal vaccine generated higher titers of PspA-specific IgG1 and nasal IgA (*Figure 1C*). Additionally, they had stronger IL-13 ($T_H2$) and IL-17 ($T_H17$) responses in the ex vivo recall assay than the 2′3′-cGAMP adjuvanted pneumococcal vaccine (*Figure 1D*). The IgG2C and IFNγ ($T_H1$) responses were similar between CDG and 2′3′-cGAMP adjuvanted vaccine (*Figure 1C,D*).

Last, we examined the protective immunity against pneumococcal infection in CDG plus PspA vs 2′3′-cGAMP plus PspA immunized mice. We found that mice immunized with CDG adjuvanted pneumococcal vaccine have a lower bacterial burden in the spleens and lungs than mice immunized with 2′3′-cGAMP adjuvant pneumococcal vaccine (*Figure 1E*). We concluded that, in mice, CDG, as a mucosal adjuvant, generated better Ag-specific Ab production as well as stronger $T_H$ responses than the mammalian cyclic dinucleotide 2′3′-cGAMP. This translated into better protection against pneumococcal infection in vivo.

## Intranasal administered CDG does not cause lung injury

Next, we examined the safety profile of CDG adjuvant. At the dose of CDG used in *Figure 1* (5 μg), we saw only very mild neutrophil infiltration in Bronchoalveolar lavage fluid (BALF) (*Figure 2A*) and lungs (*Figure 2D*). We also determined lung permeability by serum albumin level in BALF. There was no significant difference in samples from saline or CDG treated mice (*Figure 2B*). Last, lung histology also did not reveal any lung damage in CDG treated mice (*Figure 2C*). We concluded that intranasally administered CDG, at the dose used as an effective mucosal adjuvant, did not cause lung injury.

## Intranasal administered CDG does not cause excessive inflammatory responses

Next, we examined CDG induced cellular responses in vivo. Besides a mild increase in the number of neutrophils in lung, there was also a ~twofold increase in Ly6C$^{hi}$ monocytes in the lung after intranasal CDG administration (*Figure 2D*). There were no significant increases in numbers of Mast cells or eosinophils in lungs at the vaccine adjuvant dose of CDG used (5 μg) (*Figure 2D*). There were also no increases in the number of B cells or NK cells (*Figure 2D,E*).

Ly6C$^{hi}$ monocytes could differentiate into DCs, mainly CD11B$^+$ myeloid DCs, in situ. We did not find any difference in total DC number, or CD103$^+$, CD11B$^+$ DCs subset numbers in the lungs after CDG treatment (*Figure 2F,G*).

CDG induces the production of the proinflammatory cytokines TNFα and IL-1β in vitro (*Karaolis et al., 2007*). We confirmed this in vivo (*Figure 2H*). However, we found that CDG also induced potent IL-10 production, an anti-inflammatory cytokine, in vivo (*Figure 2I*). Furthermore, CDG induced strong IL-22 production in vivo (*Figure 2I*), which is important for lung epithelium repair (*Paget et al., 2012*; *Pociask et al., 2013*). The balanced production of inflammatory and anti-inflammatory cytokines by CDG likely explains the absence of excess inflammatory responses in vivo.

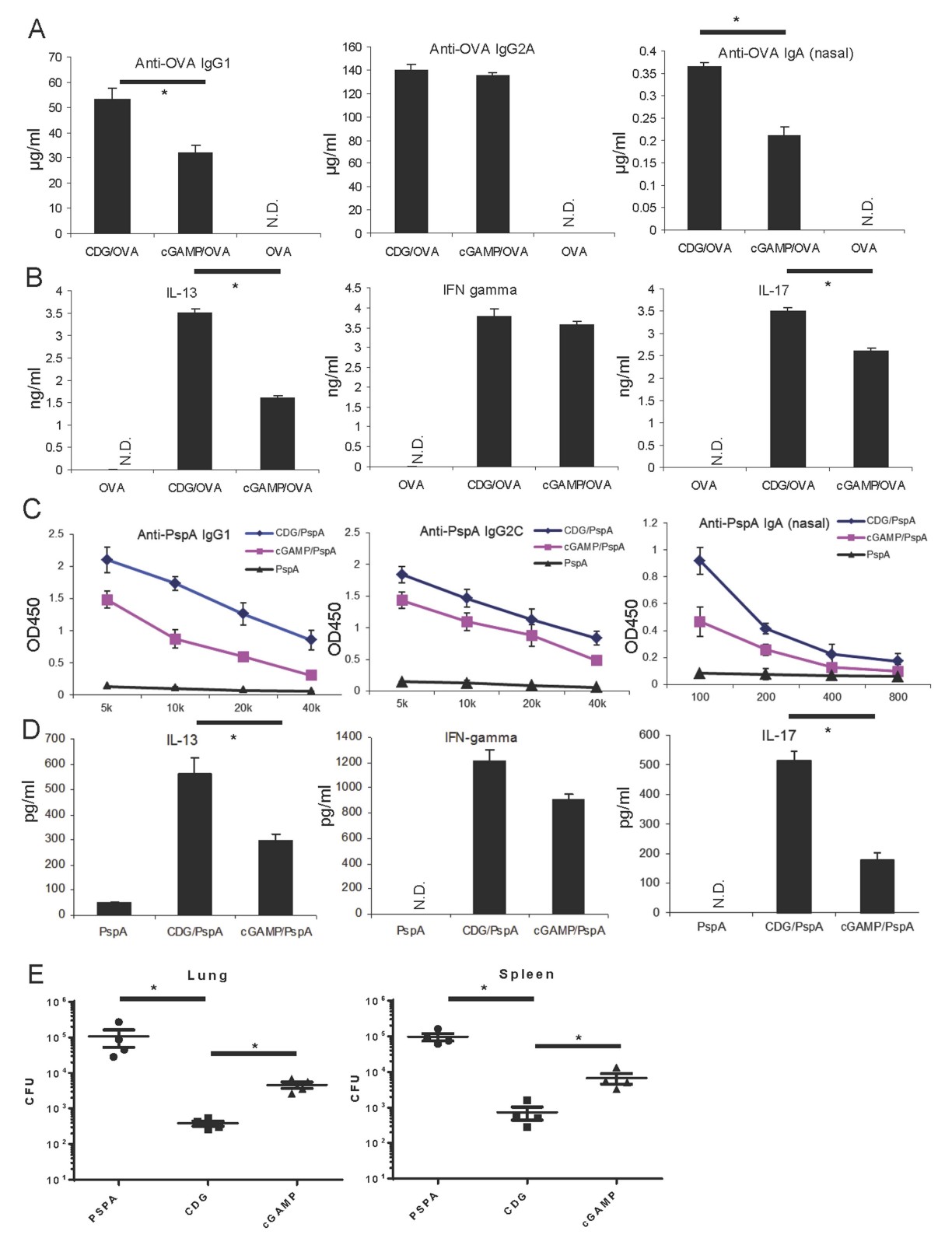

**Figure 1**. Cyclic di-GMP (CDG) is a better mucosal pneumococcal vaccine adjuvant than the mammalian cyclic dinucleotide 2′3′-cyclic GMP-AMP (cGAMP) in mice. (**A**) BALB/c mice were intranasally (i.n.) immunized with three doses (14 days apart) of OVA (20 μg) alone or together with 5 μg CDG or 5 μg 2′3′-cGAMP. Each group consisted of four mice. Sera or nasal washes from the 4 mice in the same group were pooled. Blood and nasal washes

*Figure 1. continued on next page*

Figure 1. Continued

samples were collected 14 days after the last immunization. Anti-OVA IgG1, IgG2A, and IgA were quantified by ELISA. n = 3. (**B**) Splenocytes from immunized BALB/c mice were stimulated with 50 µg/ml OVA for 4 days in culture. Supernatants from the same group were pooled together. Cytokines were measured in the supernatant by ELISA. n = 3. (**C**) C57BL/6 mice were immunized with 3 doses of pneumococcal surface protein A (PspA) (2 µg) alone or together with 5 µg CDG or 5 µg 2′3′-cGAMP as in **A**. Blood and nasal washes were collected 14 days after the last immunization. Anti-PspA IgG1, IgG2C, and IgA were measured by ELISA as in **A**. n = 3. (**D**) Splenocytes from immunized C57BL/6 mice were stimulated with 5 µg/ml PspA for 4 days in culture. Cytokines were measured in the supernatant by ELISA as in **B**. n = 3. (**E**) Immunized mice were infected (i.n.) with *S. pneumoniae* ($\sim$5.0 × 10$^6$ c.f.u.). At 48 hr post infection, lung and spleen bacterial burden were determined. n = 2. Graph present means ± standard error from three independent experiments. Significance is represented by an asterisk, where p < 0.05.

## CDG induces potent type II (IFNγ) and III IFN (IFN λ) production in vivo

While CDG-induced TNFα and IL-22 production were completely dependent on the expression of MPYS, IL-1β, and IL-10 production in vivo were only partially dependent on MPYS (*Figure 2H,I*). This was surprising considering that MPYS was the proposed direct receptor for CDG in mammalian cells. We then investigated the cytokine milieu in the lungs after CDG administration in WT and *Tmem173$^{-/-}$* mice.

We first examined the production of type I IFN, the signature cytokine stimulated by MPYS/STING, in the lungs. Although we detected low-level background IFNβ production in the lungs, CDG treatment did not increase IFNβ levels above the background (*Figure 3A*). This was consistent with our previous observation that the mucosal adjuvant activity of CDG is type I IFN independent (*Blaauboer et al., 2014*).

Surprisingly, we detected potent type III IFN (IFN λ) production in the lungs after intranasal administration of 5 µg CDG (*Figure 3A*). Type III IFN activates similar groups of interferon stimulating genes (ISGs) as type I IFN. However, their receptors are mainly expressed on lung epithelial cells (*Zhou et al., 2007*). Furthermore, neutralizing IFNλ in vivo did not affect the adjuvant activity of CDG (*Figure 3—figure supplement 1*).

We also detected strong CDG induced type II IFN (IFN γ) in vivo (*Figure 3B*). Both type II and III IFN production by CDG were absent in MPYS$^{-/-}$ mice (*Figure 3A,B*). We concluded that intranasally administered CDG, at the dose used as an effective mucosal adjuvant, induces potent type II and III IFN, but not type I IFN production in vivo.

## CDG induces T$_H$1, T$_H$2, and T$_H$17 polarizing cytokines in vivo

CDG immunization generates T$_H$1, T$_H$2, and T$_H$17 responses. Type II IFN is a T$_H$1 polarizing cytokine. We examined if CDG induced other T$_H$ polarizing cytokines in the lungs. Indeed, intranasally administered CDG induced T$_H$1 polarizing cytokine IL-12p70, T$_H$2 polarizing cytokine IL-5, to a lesser degree IL-4 and IL-13, and T$_H$17 polarizing cytokines IL-23, IL-6, and TGF-β1 (*Figure 3B–D*). Except for IL-6 production, all these CDG induced cytokines were absent in *Tmem173$^{-/-}$* mice (*Figure 3B–3D*).

## CDG induces potent lung epithelium-derived cytokines in vivo that is only partially dependent on the expression of MPYS

Lung epithelial cells generate unique cytokines when activated, and their in vivo roles in modulating immune responses have been appreciated recently (*Hallstrand et al., 2014*). We examined lung epithelium-derived cytokines during in vivo CDG activation. Indeed, CDG induced potent IL-33 and, to a lesser degree, IL-1α and TSLP production (*Figure 3E*). Distinct from many of the cytokines examined above, these CDG induced lung epithelium cytokines were only partially dependent on the expression of MPYS (*Figure 3E*).

Noticeably, all cytokines were detected at both 6 hr and 24 hr post CDG administration (*Figure 2* and *Figure 3*). In fact, we could detect these cytokines as early as 4 hr post CDG administration in vivo. The rapid production of these cytokines by CDG in vivo suggested that CDG induced cytokines were a primary response rather than a secondary effect.

## CDG generates IL-12p70 producing DC in vivo

The rapid generation of T$_H$1, T$_H$2, and T$_H$17 polarizing cytokines in the lungs from CDG treated mice led us to hypothesize that CDG directly activated pulmonary DCs in vivo that generated T$_H$ polarizing cytokines, leading to differentiated T-helper cell responses.

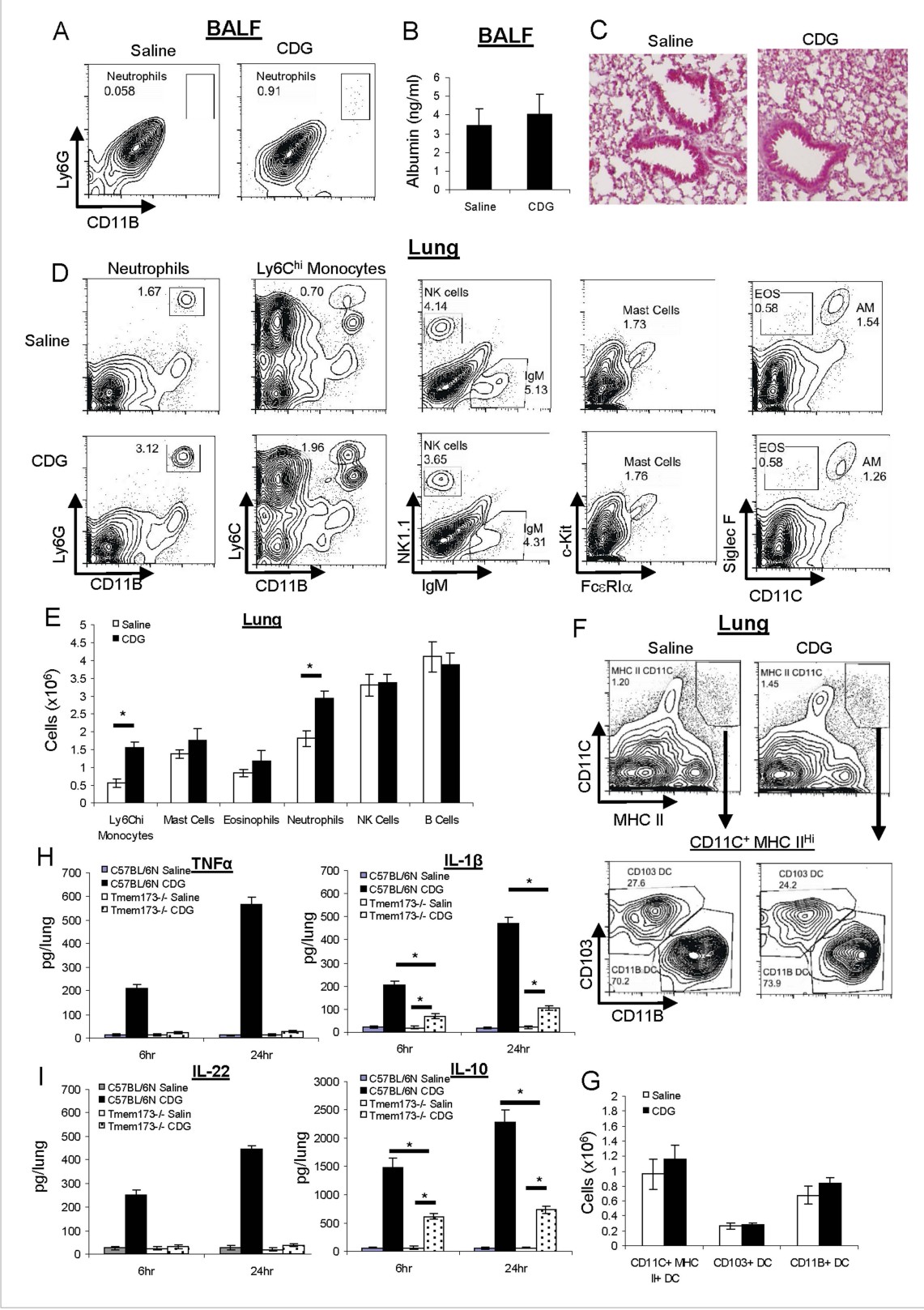

**Figure 2**. CDG does not cause lung injury or excess inflammatory responses in vivo. (**A**) C57BL/6 mice were treated (i.n.) with saline or CDG (5 µg) for 20 hr. Cells in Bronchoalveolar lavage fluid (BALF) were analyzed by FACScan with indicated Abs. Live cells were gated. n > 3. (**B**) Serum albumin level in the collected BALF was measured by ELISA (#GWB-282C17; GenWay). n = 3. (**C**) Lung sections from treated mice were fixed and histology was determined by

*Figure 2. continued on next page*

Figure 2. Continued

Hematoxylin and eosin stain. n = 3. (**D–G**) Mice were treated as in **A**. Lung cells were analyzed by FACScan with indicated Abs and quantified. Live cells were gated. n = 3. (**H–I**) C57BL/6 and MPYS$^{-/-}$ mice were treated (i.n.) with saline or CDG (5 μg) for the indicated time. Cytokines were determined in lung homogenates by ELISA. n > 3. Graph present means ± standard error from three independent experiments. Significance is represented by an asterisk, where p < 0.05.

To test this hypothesis, we performed intracellular cytokine staining in pulmonary DC from CDG treated mice. We focused on detecting $T_H1$ promoting DCs as defined by IL-12p35 or IFNγ production. Unlike IL-12p40, IL-12p35 is unique to IL-12p70. We gated MHC II$^{hi}$CD11C$^+$ DCs from total lung and looked for IL-12p35$^+$ or IFNγ$^+$ DC (*Figure 3F*). IL-12p35$^+$ DC accounted for ~0.035% of DCs, which amounted to less than 500 of these cells in a lung from a CDG treated mouse (*Figure 3G*). The percentage of IL-12p35$^+$ IFNγ$^+$ DC was ~0.01% (*Figure 3F,G*). As a control, no IL-12p35$^+$ DCs were detected in saline treated mice (*Figure 3F*).

## CDG enhances Ag uptake in APCs and non-APCs in vivo

Next, we investigated how CDG affects DCs in vivo. We used Alexa Fluor 647 conjugated OVA Ag (OVA-647) to examine Ag uptake and DQ-OVA for Ag processing (*Figure 4A,B*). DQ-OVA is a self-quenched conjugate of OVA that exhibits bright, photostable, and pH insensitive green fluorescence upon proteolytic degradation (DQ-Green) (*Figure 4A*). Furthermore, when digested fragments of DQ-OVA accumulate in organelles at a high concentration, it forms excimers emitting red fluorescence (DQ-Red) (*Figure 4A*).

We intranasally administered mice with the OVA-647 plus DQ-OVA in the presence or absence, of CDG. After 24 hr, we examined OVA-647$^+$ and DQ$^+$ cells in the lung. We found that including CDG in the immunization dramatically improved Ag uptake, as indicated by the increased number of OVA-647$^+$ cells in lung (*Figure 4C*). Furthermore, ~34% of these OVA-647$^+$ cells were DQ$^+$, which indicated that only a portion of OVA-647$^+$ cells has the ability to process Ag (*Figure 4C*). The DQ$^+$ cells included both DQ-Green and DQ-Red cells (*Figure 4C*).

Of note, the CDG induced OVA-647$^+$ cells included both MHC II$^+$ APCs and MHC II$^-$ non-APCs (*Figure 4C*). We focused on MHC II$^+$ APCs. There are three populations of antigen presenting cells (APCs) from WT mice: MHC II$^{hi}$CD11C$^+$ (i.e., DCs), MHC II$^{low}$CD11C$^+$ and MHC II$^{int}$CD11C$^-$ (*Figure 4D*). Notably, the majority of OVA-647$^+$MHC II$^{low}$CD11C$^+$ cells were OVA-647$^{hi}$ cells, while the majority of OVA-647$^+$MHC II$^{hi}$CD11C$^+$ and OVA-647$^+$ MHC II$^{int}$CD11C$^-$ cells were OVA-647$^{low}$ cells (*Figure 4F*). A previous study established that OVA-647$^{hi}$ cells were generated via receptor-mediated endocytosis while OVA-647$^{low}$ cells were a result of pinocytosis-mediated Ag uptake (*Burgdorf et al., 2007*). Thus, CDG predominantly enhanced receptor-mediated endocytosis in MHC II$^{low}$CD11C$^+$ and pinocytosis in MHC II$^{hi}$CD11C$^+$ and MHC II$^{int}$CD11C$^-$ cells.

## MHC II$^{hi}$CD11C$^+$ and MHC II$^{int}$CD11C$^-$ cells are activated by CDG in vivo

CDG treatment activates cells in vitro, which depends on MPYS (*Jin et al., 2011a*; *Pociask et al., 2013*). We next wanted to know which APCs were activated during intranasal administration of CDG. APCs increase CD86 expression during activation. In the OVA-647$^+$ MHC II$^{low}$CD11C$^+$ population, there was no increase of the activation marker CD86 (*Figure 4E*). In the remaining two APC populations, MHC II$^{hi}$CD11C$^+$ and MHC II$^{int}$CD11C$^-$, the OVA-647$^+$ cells had increased CD86 expression (*Figure 4E*). Thus, CDG activates MHC II$^{hi}$CD11C$^+$ and MHC II$^{int}$CD11C$^-$, but not MHC II$^{low}$CD11C$^+$ APCs in vivo. The total numbers of CD86$^+$MHC II$^{hi}$ activated OVA-647$^+$ cells were similar between MHC II$^{hi}$CD11C$^+$ and MHC II$^{int}$CD11C$^-$ APCs (*Figure 4G*).

Of note, while CDG selectively activated different APCs, it did enhance Ag uptake in all three APCs populations in vivo (*Figure 4D*). This suggested that cell activation is not a prerequisite for CDG enhanced Ag uptake in vivo.

## CDG enhances Ag processing in APCs in vivo

CDG also dramatically increased numbers of DQ$^+$ cells in vivo (*Figure 5A*). As shown in *Figure 4C*, only a third of OVA-647$^+$ cells were able to process Ag (DQ$^+$). We, thus, focused on DQ$^+$ cells, where

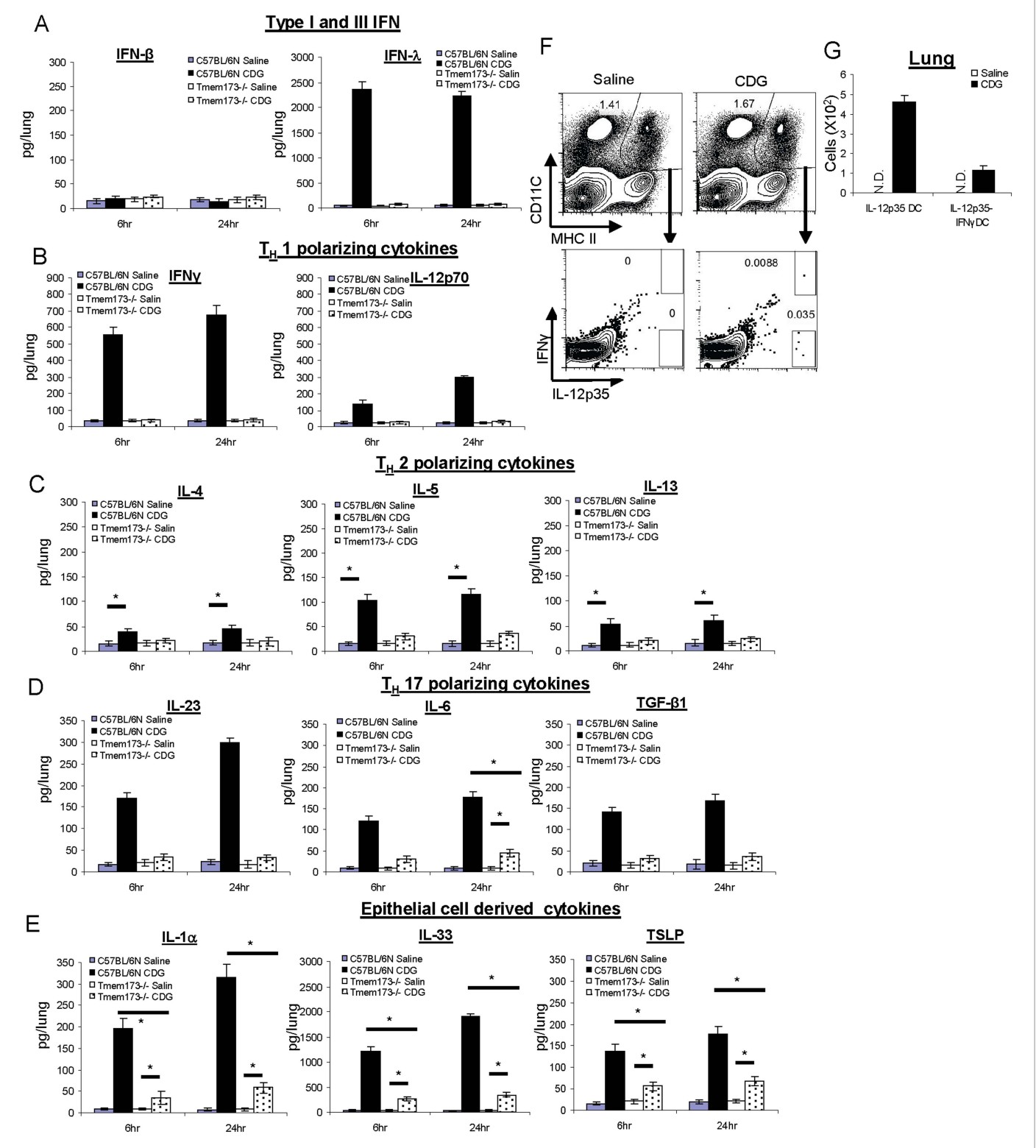

**Figure 3**. CDG induces a variety of cytokines in lung that is dependent on the expression of MPYS. (**A–E**) C57BL/6 and *Tmem173⁻/⁻* mice were treated (i.n.) with saline or CDG (5 µg) for the indicated time. Cytokines were determined in lung homogenates by ELISA. n > 3. (**F–G**) C57BL/6 mice were treated (i.n.) with saline or CDG (5 µg) for 5 hr. IL-12-p35 and IFNγ positive dendritic cells (DCs) were identified by intracellular cytokine stains and

*Figure 3. Continued*

quantified. n = 3. Graph present means ± standard error from three independent experiments. Significance is represented by an asterisk, where p < 0.05.

The following figure supplement is available for figure 3:

**Figure supplement 1**. IFNλ production is dispensable for the mucosal adjuvant activity of CDG.

Ag was processed. Gated on the DQ$^+$ lung cells, we found that the vast majority of DQ$^+$ cells (∼94%) were OVA-647$^+$ cells (*Figure 5A*). Since cells have to take up Ag (OVA-647$^+$) before processing it (DQ$^+$), the small percentage of DQ$^+$OVA-647$^-$ cells (∼5%) could represent cells that lost the OVA-647 signal during the Ag process. Alternatively, DQ-OVA signal could be more sensitive than the OVA-647 signal.

The DQ$^+$OVA$^+$ consisted of two populations: OVA-647$^{hi}$ and OVA-647$^{low}$ cells (*Figure 5A*). OVA-647$^{hi}$ cells were generated via receptor-mediated endocytosis while OVA-647$^{low}$ cells were a result of pinocytosis-mediated Ag uptake (*Burgdorf et al., 2007*). The DQ$^+$OVA-647$^{hi}$ cells had a strong DQ-Red signal, indicating that processed Ag concentration was high in these cells (*Figure 5B*). The DQ$^+$ OVA-647$^{low}$ cells were DQ-Red negative, though they still processed Ag as they were DQ-Green$^+$ (*Figure 5B*). Thus, the receptor-mediated Ag endocytosis generates DQ-Green$^+$DQ-Red$^+$ cells, while pinocytosis-mediated Ag uptake generates DQ-Green$^+$DQ-Red$^-$ cells.

## DQ$^+$ lung cells are APCs

We found that the DQ$^+$ cells were almost exclusively APCs (MHC II$^+$ cells) (*Figure 5C*). This was different from the OVA-647$^+$ cells, which included both APC and non APCs (*Figure 4C*). Furthermore, the vast majority of the DQ$^+$ lung cells (>90% of DQ$^+$ cells) were MHC II$^{low}$CD11C$^+$ APC (*Figure 5C*). The MHC II$^{hi}$CD11C$^+$ and MHC II$^{int}$ CD11C$^-$ APCs accounted for ∼1% and 2% of DQ$^+$ cells, respectively (*Figure 5C*). The MHC II$^{low}$CD11C$^+$DQ$^+$ cells were Siglec F$^+$ (*Figure 5C*) cells, which should be characterized as alveolar macrophages. This suggested that alveolar macrophages are the dominant Ag uptake and processing cells during intranasal CDG administration. Indeed, ∼26% of total lung Siglec F$^+$ alveolar macrophages were DQ$^+$ cells (*Figure 5D*). In comparison, only ∼1% of total lung cells were DQ$^+$ cells (*Figure 5A*).

## CDG administration generates mature DCs (DQ$^+$MHC II$^{hi}$CD11C$^+$)

We then investigated which DQ$^+$ APCs were activated by CDG in vivo. Studies done in OVA-647$^+$ cells revealed that OVA-647$^+$ MHC II$^{low}$CD11C$^+$ cells were not activated (*Figure 4E*). Only OVA-647$^+$MHC II$^{hi}$CD11C$^+$ and OVA-647$^+$MHC II$^{int}$CD11C$^-$ APCs were activated (*Figure 4E*). However, in DQ$^+$ cells, the only CD86$^+$ APC was MHC II$^{hi}$ DCs (*Figure 5E*). These cells also had increased CD80 expression (*Figure 5E*).

## The CD86$^+$CD80$^+$DQ$^+$ DCs are DQ-Red$^-$ pinocytosis-efficient DCs

CDG predominantly activated OVA-647$^{low}$ APCs (*Figure 4E,F*), which took up Ag via pinocytosis. We found that OVA-647$^{low}$ cells were all DQ-Red$^-$ while OVA-647$^{hi}$ cells were all DQ-Red$^+$ (*Figure 5B*). Similarly, the vast majority of MHC II$^{hi}$CD86$^+$CD80$^+$DQ$^+$ cells were DQ-Red$^-$ cells (*Figure 5F*). We concluded that during intranasal administration of CDG, the only APCs that took up Ag (OVA-647$^+$), processed Ag (DQ-Green$^+$) and activated (CD86$^+$CD80$^+$), were MHC II$^{hi}$ pinocytosis-efficient (DQ-Red$^-$) DCs.

CDG is a 690 Da small molecule with two phosphate groups that cannot directly cross cell membrane (*McWhirter et al., 2009*; *Chen et al., 2010*). Thus, during intranasal administration, CDG is likely brought into the cytosol by pinocytosis, and stimulates DCs.

## CDG enhances Ag uptake, processing, and cell activation in both CD103$^+$ and CD11B$^+$ pulmonary DCs in vivo

Pulmonary DCs include CD103$^+$DCs and CD11B$^+$DCs. By co-administration of DQ-OVA and CDG, we found that CDG enhanced Ag uptake and processing, as indicated by increased numbers of DQ$^+$ cells,

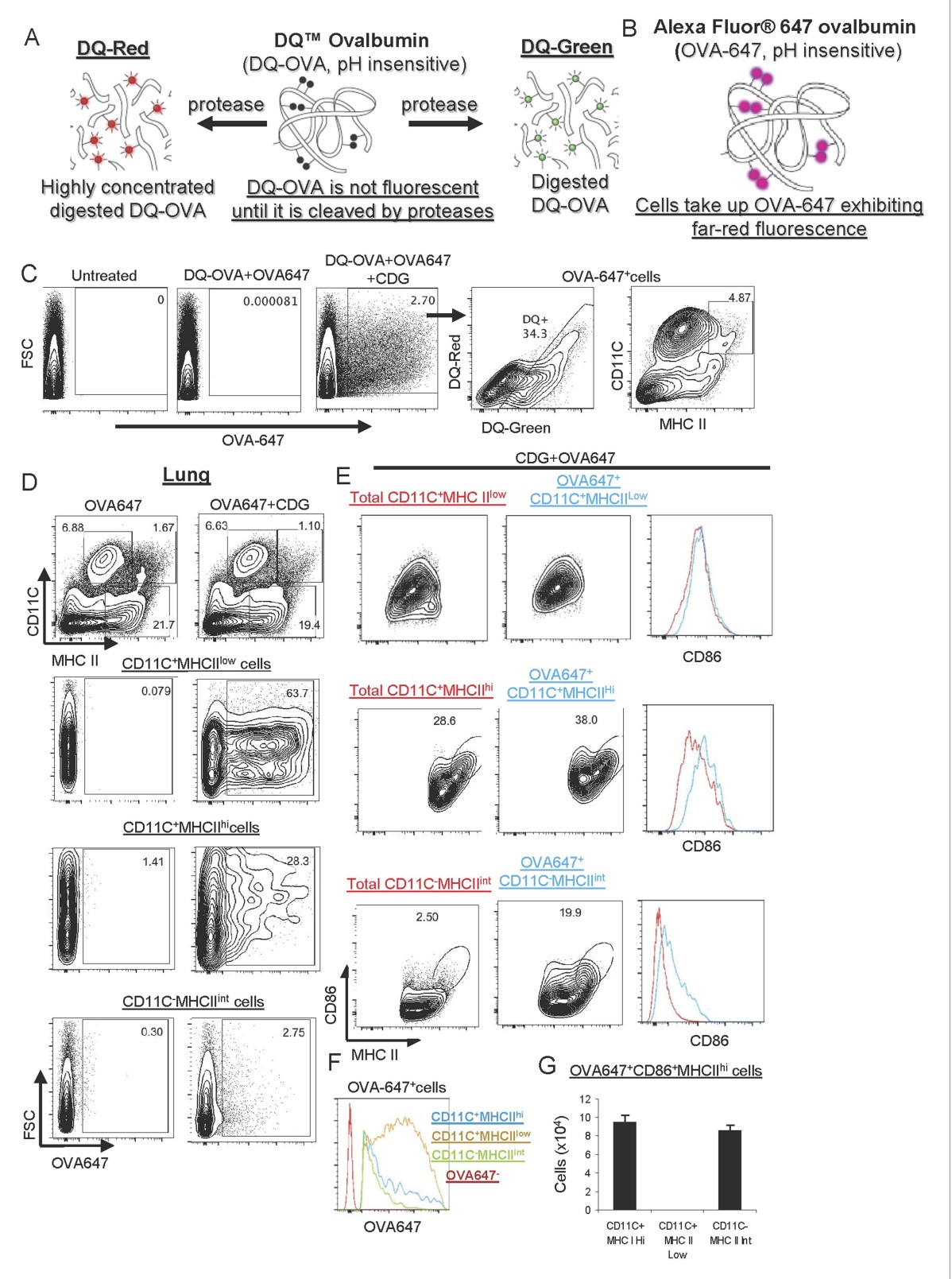

**Figure 4**. CDG enhances Ag uptake and activates pinocytosis-efficient antigen presenting cells (APCs) in vivo. (**A–B**) A cartoon showing mechanism of action of DQ-OVA (**A**) and OVA-647 (**B**). (**C**) Flow cytometry analysis of lung cells from C57BL/6 mice treated with saline, DQ-OVA(20 µg) + OVA-647(20 µg) or 5 µg CDG + DQ-OVA(20 µg) + OVA-647(20 µg) for 20 hr. Live cells were gated. n > 3. (**D**) Flow cytometry analysis of lung cells from C57BL/6 mice

*Figure 4. continued on next page*

Figure 4. Continued
treated with OVA-647(20 μg) or 5 μg CDG + OVA-647(20 μg) for 20 hr. Live cells were gated. n > 3. (E) Flow cytometry analysis of lung APCs from C57BL/6 mice treated with 5 μg CDG + OVA-647(20 μg) for 20 hr. Live cells were gated. n > 3. (F) Histogram of OVA-647 signals from OVA-647+ APCs. n > 3. (G) Cell numbers of activated OVA-647+ APCs were quantified. n > 3. Graph present means ± standard error from three independent experiments.

in both CD103+ and CD11B+ DCs (*Figure 6A,D*). We did notice that CD103+DCs had a higher percentage of DQ+ cells than the CD11B+ DCs (*Figure 6A,D*). Both DC subsets had DQ-Red+ and DQ-Red− populations (*Figure 6A,D*).

## CDG activates and mobilizes pulmonary CD103+ DC in vivo

Activated DCs express high MHC II and co-stimulator factor CD86. Furthermore, they migrate to draining lymph nodes (DLN), where they encounter naïve T cells and stimulate diversified T cell responses. We first examined the actions of CD103+ DC. A significant portion of lung CD103+DQ+ DCs (∼35%) from CDG treated mice were MHC IIhiCD86+ activated DCs (*Figure 6B*). The absolute number of CD103+DQ+CD86+ cells was also recorded (*Figure 6G*). Interestingly, these activated DQ+ CD103+CD86+ DCs were all DQ-Red negative cells (*Figure 6B*). In fact, it appeared that all DQ-Red− cells were CD86+ DCs and all DQ-Red+ cells were CD86− (*Figure 6B,C*).

We then examined migratory CD103+ DC in lung DLN. CDG treatment increased total CD103+ DCs numbers in DLN (*Figure 6H,L*). However, only a very small percentage of the migratory

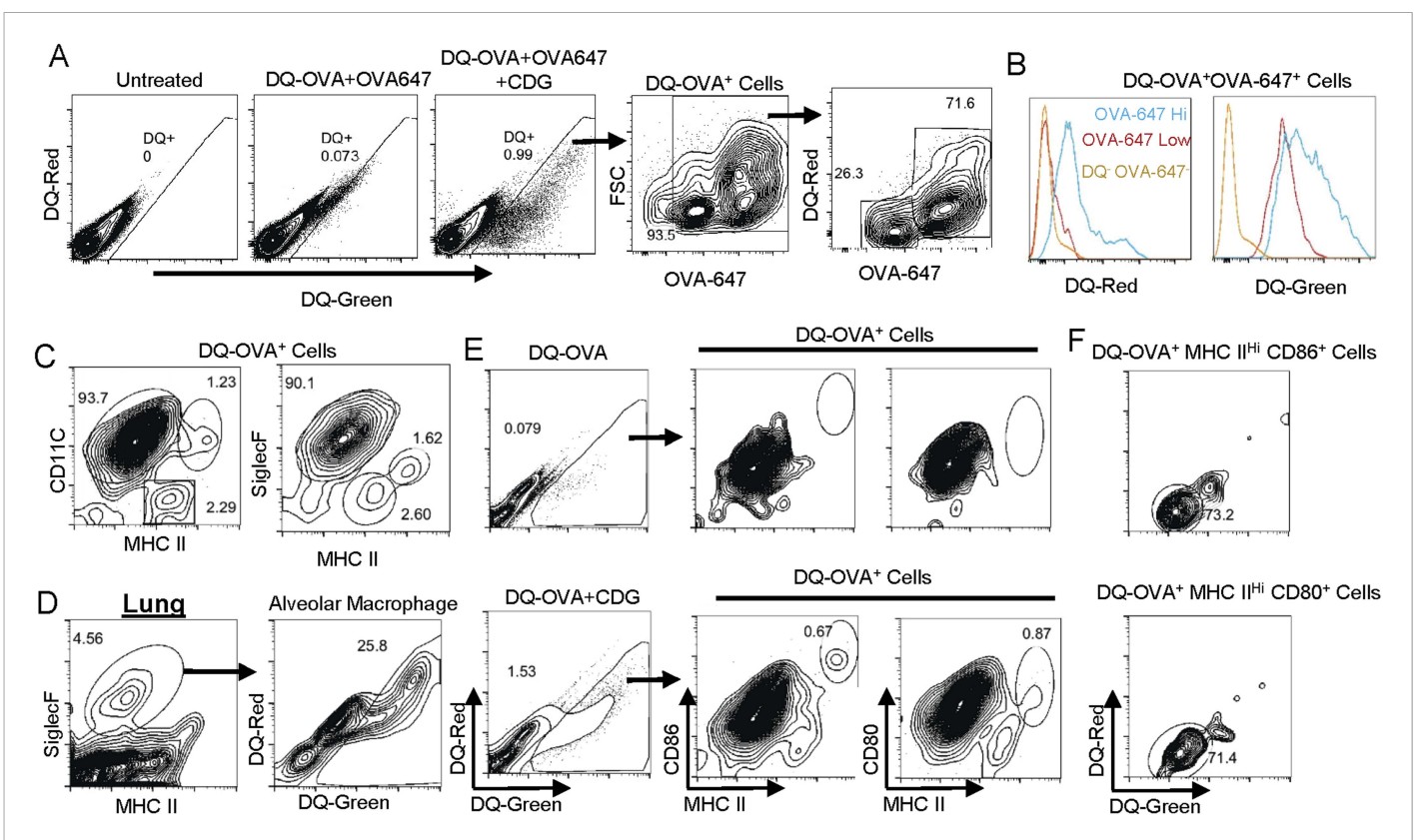

**Figure 5**. CDG generates mature DCs in vivo. (**A**) Flow cytometry analysis of lung cells from C57BL/6 mice treated with saline, DQ-OVA(20 μg) + OVA-647 (20 μg) or 5 μg CDG + DQ-OVA(20 μg) + OVA-647(20 μg). Live cells were gated. n > 3. (**B**) Histogram of DQ-Red and DQ-Green signals from indicated populations. n > 3. (**C**) Flow cytometry analysis of DQ+ lung cells from CDG + DQ-OVA treated (i.n.) C57BL/6 mice. Live DQ+ cells were gated. n > 3. (**D**) Flow cytometry analysis of lung cells from CDG + DQ-OVA treated (i.n.) C57BL/6 mice. Live cells were gated. n = 3. (**E**) Flow cytometry analysis of DQ+ lung cells from DQ-OVA or CDG + DQ-OVA treated (i.n.) C57BL/6 mice. Live were gated. n > 3. (**F**) Flow cytometry analysis of the indicated population from lung cells of CDG + DQ-OVA treated (i.n.) C57BL/6 mice. Gated on live DQ+ CD80+MHC II+ or live DQ+ CD86+MHC II+ cells. n > 3.

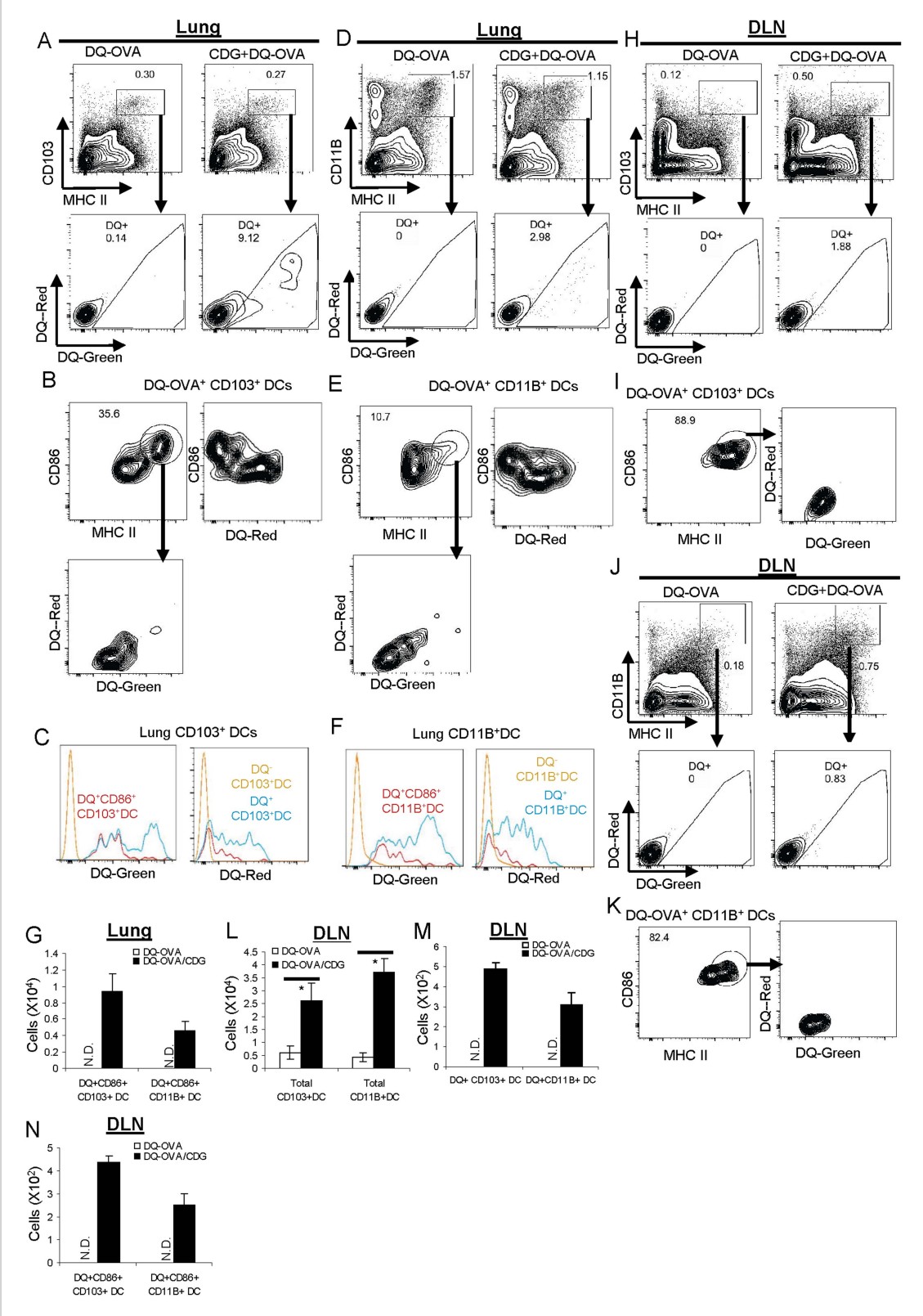

**Figure 6**. CDG activates pinocytosis-efficient CD103+ and CD11B+ DCs in vivo. (**A** and **D**) Flow cytometry analysis of lung cells from C57BL/6 mice treated (i.n.) with DQ-OVA (20 µg) or CDG (5 µg) + DQ-OVA (20 µg) for 20 hr. Live cells were gated. n > 3. (**B** and **E**) Flow cytometry analysis of DQ+ DCs from lung of CDG + DQ-OVA treated C57BL/6 mice. Cells were gated on live DQ+CD103+MHC II+ or live DQ+CD11B+MHC II+ cells. n > 3. (**C** and **F**) Histogram of
*Figure 6. continued on next page*

*Figure 6. Continued*

DQ-Green and DQ-Red signals from cell populations in **B** and **E**. n > 3. (**G**) Total cell number of CD86$^+$DQ$^+$ DCs in lung. n = 3. (**H** and **J**) Flow cytometry analysis of lung draining lymph nodes (DLN) from DQ-OVA or CDG + DQ-OVA treated C57BL/6 mice. Live cells were gated. n = 3. (**I** and **K**) Flow cytometry analysis of DQ$^+$ DCs in DLN of CDG + DQ-OVA treated C57BL/6 mice. Cells were gated on live DQ$^+$CD103$^+$MHC II$^+$ or live DQ$^+$CD11B$^+$MHC II$^+$ cells. n = 3. (**L–N**) Total cell numbers of DCs, DQ$^+$DCs, and CD86$^+$DQ$^+$DCs in DLN from DQ-OVA or CDG + DQ-OVA treated mice. n = 3. Graph present means ± standard error from three independent experiments. Significance is represented by an asterisk, where p < 0.05.

CD103$^+$ DCs (∼1.8%) were DQ$^+$ (*Figure 6H,M*). This indicated that a large portion of CD103$^+$ DCs were migratory, likely activated by CDG, but did not take up the DQ-OVA Ag. Among those DQ$^+$ CD103$^+$ migratory DCs, the vast majority of them were MHC II$^{hi}$CD86$^+$ cells (∼89%) (*Figure 6I*), which indicated that these migratory DQ$^+$CD103$^+$DCs were indeed activated DCs. Consistent with the finding in the lungs, all these migratory DQ$^+$CD103$^+$ in DLN were DQ-Red$^-$ cells (*Figure 6I*). The number of DQ$^+$ CD86$^+$ migratory CD103DCs was recorded (*Figure 6N*).

## CDG activates and mobilizes pulmonary CD11B$^+$ DC in vivo

We next examined the activation of CD11B$^+$ DCs by CDG in vivo. Similar to CD103$^+$DCs, we found that (1) a portion of lung CD11B$^+$DQ$^+$ DCs were MHC II$^{hi}$CD86$^+$ activated DCs (*Figure 6E*); (2) these activated DQ$^+$CD11B$^+$CD86$^+$ DCs were all DQ-Red negative cells (*Figure 6E*); (3) all the DQ-Red$^-$DC-Green$^+$CD11B$^+$ DCs were CD86$^+$ (*Figure 6E*); (4) total CD11B$^+$ DCs numbers were increased in DLN (*Figure 6J,L*); (5) only a very small percentage of these CD11B$^+$ DCs (∼0.8%) were DQ$^+$ (*Figure 6J,M*); (6) the vast majority of DQ$^+$CD11B$^+$ migratory DC were MHC II$^{hi}$CD86$^+$ cells (∼82%) (*Figure 6K,N*); (7) all these migratory DQ$^+$CD11B$^+$ were DQ-Red$^-$ cells (*Figure 6K*).

## CDG differentially mobilizes pulmonary DC in vivo based upon their endocytosis ability

Our investigation, so far, revealed that CDG differentially mobilized two major types of Ag-loaded pulmonary DCs: DQ-Green$^+$DQ-Red$^-$CD86$^+$ and DQ-Green$^+$DQ-Red$^+$CD86$^-$ DCs. DQ-Red$^-$ DQ-Green$^+$ cells represented pinocytosis-efficient DCs (*Figure 6A,B*). The fact that these were the only CD86$^+$ and DQ$^+$ migratory DCs found in DLN after CDG treatment suggested that CDG only activated pinocytosis-efficient DCs in vivo. It did not matter whether they were CD103$^+$ or CD11B$^+$ DCs (*Figure 6H,J*).

In contrast, all the DQ-Green$^+$DQ-Red$^+$ cells were CD86$^-$ and non-migratory, suggesting that though CDG enhanced Ag uptake in these cells (*Figure 4*), it did not lead to cell activation. It further strengthened the notion that activation of these cells is not a prerequisite for CDG enhanced Ag uptake (*Figure 4E*).

## MPYS is critical for CDG enhanced Ag uptake in vivo

Mucosal adjuvant activity of CDG requires MPYS in vivo (*Blaauboer et al., 2014*). We next asked how MPYS regulated CDG enhanced Ag uptake and processing in vivo. Upon co-administration of OVA-647 and CDG, lung cells from *Tmem173$^{-/-}$* mice had no increased OVA-647$^+$ cells (*Figure 7A*).

Since DQ-OVA may be more sensitive than OVA-647 in detecting Ag-loaded APCs (*Figure 5A*), we examined DQ-OVA signals in lung cells from CDG treated *Tmem173$^{-/-}$* mice. As expected, *Tmem173$^{-/-}$* mice had significantly less CDG induced DQ-OVA$^+$ cells than WT mice (*Figure 7B*).

Both CD103$^+$DC (*Figure 7C*) and CD11B$^+$ DC (*Figure 7D*) from CDG treated *Tmem173$^{-/-}$* mice, had dramatically decreased DQ$^+$ cells (*Figure 7E*). This included both the DQ-Green$^+$DQ-Red$^+$ receptor-mediated endocytosis and DQ-Green$^+$ DQ-Red$^-$ pinocytosis cells. We concluded that MPYS is critical for CDG induced DC Ag endocytosis and pinocytosis in vivo.

## MPYS is critical for the generation of activated DQ$^+$ DCs by CDG in vivo

We next examined activated DQ$^+$ DCs in *Tmem173$^{-/-}$* mice. As expected, no CD86$^+$CD80$^+$DQ$^+$ MHC II$^{hi}$ cells can be detected in CDG treated *Tmem173$^{-/-}$* mice (*Figure 7F*). This was consistent with the finding that *Tmem173$^{-/-}$* mice had a severe defect on CDG induced cytokine production in vivo (*Figure 3*). CDG is likely brought into cells by pinocytosis in vivo (*Figures 4E, 5F, 6B,E*) and MPYS is critical for CDG enhanced pinocytosis (*Figure 7A*). Thus, the reasons for the lack of overall activation

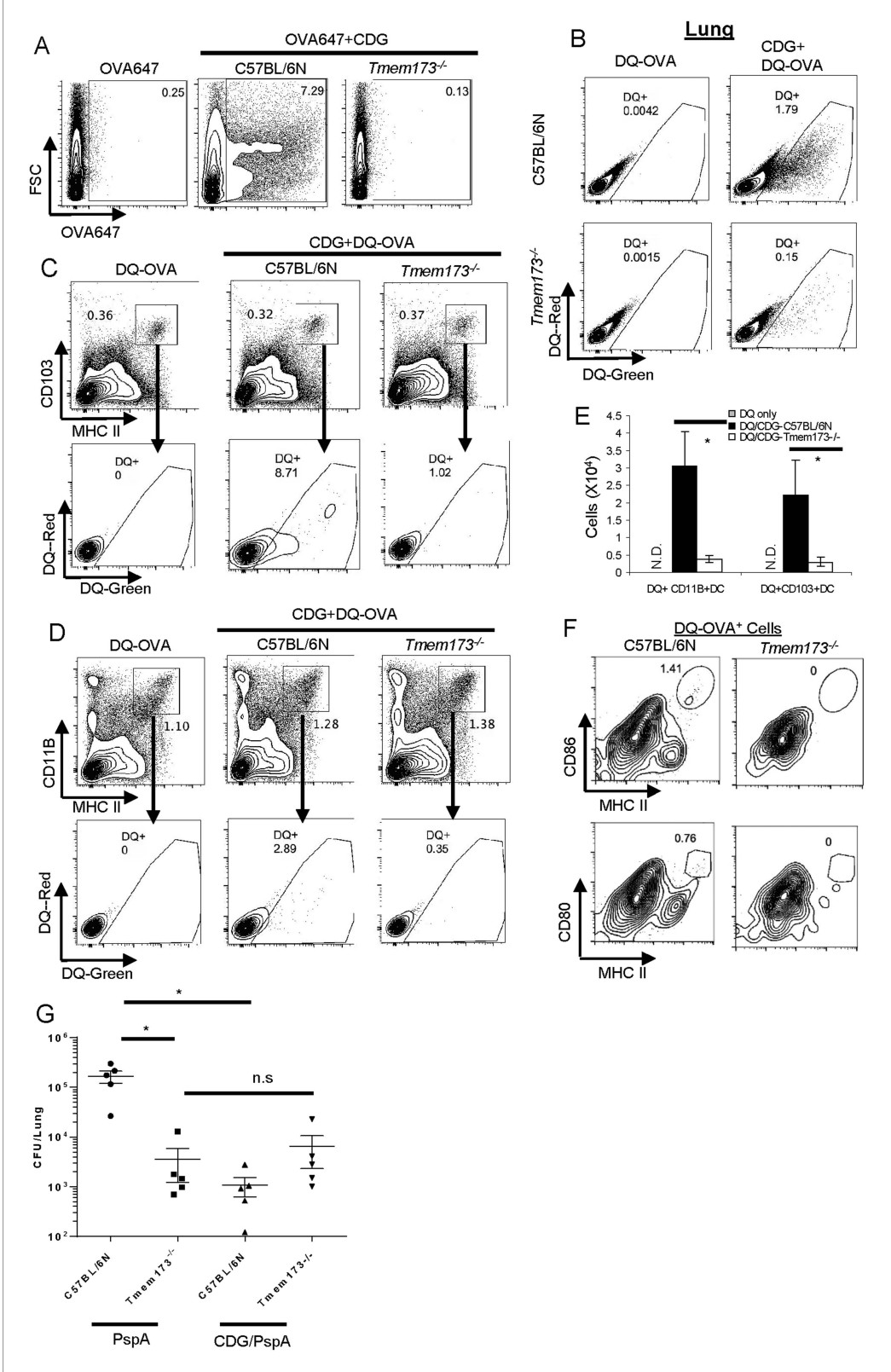

**Figure 7**. MPYS is critical for CDG induced Ag uptake and activation in vivo. (**A**) Flow cytometry analysis of lung cells from OVA-647(20 μg) or OVA-647(20 μg) + CDG(5 μg) treated (i.n.) C57BL/6 or *Tmem173*[−/−] mice. Live cells were gated. n > 3. (**B**) Flow cytometry analysis of lung cells from DQ-OVA (20 μg) or DQ-OVA(20 μg) + CDG(5 μg) treated
*Figure 7. continued on next page*

*Figure 7. Continued*

(i.n.) C57BL/6 or *Tmem173⁻/⁻* mice. Live cells were gated. n > 3. (**C** and **D**) Flow cytometry analysis of lung cells from C57BL/6 or *Tmem173⁻/⁻* mice treated with saline, DQ-OVA(20 µg) or 5 µg CDG + DQ-OVA(20 µg). Live cells were gated. n > 3. (**E**) DQ⁺CD103⁺DCs and DQ⁺CD11B⁺DCs numbers from DQ-OVA or CDG + DQ-OVA treated C57BL/6 and *Tmem173⁻/⁻* mice. n = 3. (**F**) Flow cytometry analysis of DQ⁺ lung cells from DQ-OVA(20 µg) + CDG(5 µg) treated (i.n.) C57BL/6 or *Tmem173⁻/⁻* mice. Live DQ⁺ cells were gated. n > 3. (**G**) One month after the last immunization, CDG/PspA or PspA immunized WT and *Tmem173⁻/⁻* mice were infected (i.n.) with *S. pneumoniae* (D39 strain, ~5.0 × 10⁶ c.f.u.). At 48 hr post infection, lung bacterial burden was determined. n = 2. Graph present means ± standard error from three independent experiments. Significance is represented by an asterisk, where p < 0.05.

by CDG in *Tmem173⁻/⁻* mice could be twofold. On one hand, CDG cannot efficiently get into MPYS-deficient cells by pinocytosis; on the other hand, the small amount of CDG that does get in cannot activate MPYS-deficient cells.

## CDG/PspA immunization did not induce protective immunity in *Tmem173⁻/⁻* mice

Next, we examined CDG/PspA vaccine induced protective immunity in the *Tmem173⁻/⁻* mice. CDG/PspA immunization significantly lowered the lung bacterial burden in the WT mice (*Figure 7G*). However, the bacterial burden in lungs from CDG/PspA and PspA immunized *Tmem173⁻/⁻* were not significantly different (*Figure 7G*). We concluded that the mucosal pneumococcal vaccine adjuvant activity of CDG requires MPYS.

Interestingly, PspA immunized *Tmem173⁻/⁻* mice had significantly lower lung bacterial burden than the PspA immunized WT mice (*Figure 7G*). We further found that *Tmem173⁻/⁻* mice, without PspA immunization, are much more resistant to *Streptococcus pneumoniae* infection than the WT mice (unpublished data). Currently, we are dissecting the in vivo mechanism underlying this MPYS-mediated susceptibility to *S. pneumoniae* infection.

## Generation of *Itgax^Cre^Tmem173^Flox/Flox^* mice

Our investigation revealed two mechanisms by which CDG promotes its adjuvant activity in vivo: (1) enhances Ag uptake in vivo; (2) activates and mobilizes DCs in vivo, specifically, the pinocytosis-efficient DQ-Green⁺DQ-Red⁻ DCs. MPYS expression is required for both actions. We then asked whether this MPYS requirement was DC-intrinsic. To achieve that, we generated *Itgax^Cre^Tmem173^Flox/Flox^* mice (*Figure 8—figure supplement 1*). Since essentially all DQ⁺ (Ag-processing) cells were CD11C⁺ (*Figure 5C*), the *Itgax^Cre^Tmem173^Flox/Flox^* mice will eliminate MPYS expression in the vast majority of DQ⁺ cells except for the CD11C⁻MHC II^int^ APCs, which accounts for ~2% of DQ⁺ cells (*Figure 5C*).

We detected MPYS expression by Flow cytometry intracellular staining. We used the same type of cell from *Tmem173⁻/⁻* mice as a negative control and the same type of cell from WT mice as a positive control. BALF cells, which are overwhelmingly CD11C^hi^ alveolar macrophages, had dramatically decreased MPYS expression (>90%) in *Itgax^Cre^Tmem173^Flox/Flox^* mice (*Figure 8—figure supplement 1B*). MPYS expression in spleen B cells (IgD⁺) or T cells (CD4⁺ or CD8⁺) did not change in *Itgax^Cre^Tmem173^Flox/Flox^* mice (*Figure 8—figure supplement 1C*).

There were two major CD11C^hi^ populations in lung cells: CD11C⁺MHC II^low^ and CD11C⁺MHC II^hi^ (*Figure 8—figure supplement 1D*). MPYS expression was dramatically decreased in both populations in *Itgax^Cre^Tmem173^Flox/Flox^* mice (*Figure 8—figure supplement 1D*). When we separated the DC population (CD11C⁺MHC II^hi^) into CD103⁺ and CD11B⁺ DCs, we found that MPYS expression was eliminated in both DCs subsets (*Figure 8—figure supplement 1D*).

The MPYS expression was down ~40% in MHC II^int^ CD11C⁻ cells from *Itgax^Cre^Tmem173^Flox/Flox^* mice (*Figure 8—figure supplement 1D*). NK cells also showed ~40% decreased MPYS expression in *Itgax^Cre^Tmem173^Flox/Flox^* mice (*Figure 8—figure supplement 1E*). We wanted to see if the partial decrease of MPYS expression affected CDG adjuvant activity. MPYS heterozygous mice, which had ~50% decreased MPYS expression (*Figure 8—figure supplement 1F*), were immunized with CDG and OVA. The total OVA-specific IgG production was similar in WT as in the *Tmem173⁺/⁻* mice (*Figure 8—figure supplement 1G*). Thus, the partial decreased MPYS expression (~50%) does not affect CDG adjuvant activity.

## CDG induced Ag uptake requires MPYS expression in CD11C$^+$ cells

We then examined CDG generated DQ-OVA$^+$ cells in the lungs of *Itgax$^{Cre}$Tmem173$^{Flox/Flox}$* mice. The total numbers of DQ$^+$ cells were dramatically decreased in *Itgax$^{Cre}$Tmem173$^{Flox/Flox}$* mice (*Figure 8A,B*). The decreased number of DQ$^+$ cells was seen in both CD103$^+$DC (*Figure 8C,D*) and CD11B$^+$ DC (*Figure 8C,E*) from *Itgax$^{Cre}$Tmem173$^{Flox/Flox}$* mice. The numbers of decreased DQ$^+$ DCs in *Itgax$^{Cre}$Tmem173$^{Flox/Flox/flox}$* was comparable to that of *Tmem173$^{-/-}$* mice (*Figure 8B*). Thus, CDG induced DC Ag uptake requires MPYS expression in CD11C$^+$ cells.

## CDG induced cytokine productions in lung requires MPYS expression in CD11C$^+$ cells

Intranasally administered CDG generated a lung cytokine milieu that is dependent on the expression of MPYS (*Figure 3*). We then examined the cytokine milieu in *Itgax$^{Cre}$Tmem173$^{Flox/Flox}$* mice. CDG induced T$_H$1 polarizing (IL-12p70 and IFN$\gamma$) and T$_H$17 polarizing (IL-23 and IL-6) cytokines were significantly decreased in *Itgax$^{Cre}$Tmem173$^{Flox/Flox}$* mice (*Figure 8F,H*). Surprisingly, we did not see much of a decrease in the T$_H$2 polarizing cytokines (IL-5, IL-13) (*Figure 8G*). Thus, MPYS expression in CD11C$^+$ cells is critical for T$_H$1 and T$_H$17 polarizing cytokine production in vivo.

CDG induced IFN-$\lambda$, IL-22, TNF-$\alpha$, and IL-1$\beta$ productions were also dramatically decreased in *Itgax$^{Cre}$Tmem173$^{Flox/Flox}$* mice (*Figure 8I,J*). Since the CD11C$^+$MHC II$^{hi}$ DCs were the only activated CD11C$^+$ cells by CDG in vivo (*Figures 4E, 5E*), we concluded that DCs expression of MPYS was critical for the generation of T$_H$1 and T$_H$17 polarizing cytokine during intranasal administration of CDG.

The lung epithelial cytokine TSLP was slightly lower in CDG treated *Itgax$^{Cre}$Tmem173$^{Flox/Flox}$* mice than in the *Tmem173$^{Flox/Flox}$* mice, but it was not statistically significant (*Figure 8K*). However, the lung epithelial cytokine IL-33 production was significantly lower in CDG treated *Itgax$^{Cre}$Tmem173$^{Flox/Flox}$* mice than in the *Tmem173$^{Flox/Flox}$* (*Figure 8K*). We favored the idea that there is a crosstalk/communication between lung epithelial cells and CD11C$^+$ cells during CDG induced immune response.

## *Itgax$^{Cre}$Tmem173$^{Flox/Flox}$* mice had impaired Ab responses to CDG adjuvanted vaccine

The *Itgax$^{Cre}$Tmem173$^{Flox/Flox}$* mice are defective in CDG induced DCs Ag uptake and activation in vivo. To determine if these mice were defective in CDG adjuvanted immune responses, we immunized these mice with CDG plus OVA and measured anti-OVA Ab productions. *Itgax$^{Cre}$Tmem173$^{Flox/Flox}$* mice exhibited significantly decreased production of anti-OVA IgG1, IgG2C, and nasal IgA (*Figure 9A*). Noticeably different from the *Tmem173$^{-/-}$* mice, where no anti-OVA Ab could be detected, CDG/OVA immunized *Itgax$^{Cre}$Tmem173$^{Flox/Flox}$* mice still generated decent amounts of anti-OVA Ab (*Figure 9A*).

We next immunized *Itgax$^{Cre}$Tmem173$^{Flox/Flox}$* mice with pneumococcal vaccine consisting of CDG and PspA and examined their Ab responses. Again, *Itgax$^{Cre}$Tmem173$^{Flox/Flox}$* mice showed decreased anti-PspA IgG1, IgG2C, and nasal IgA production in comparison to the immunized WT mice (*Figure 9B*). Similar to the CDG/OVA immunization, *Itgax$^{Cre}$Tmem173$^{Flox/Flox}$* mice still had elevated anti-PspA Ab responses compared to the MPYS$^{-/-}$ mice (*Figure 9B*).

## *Itgax$^{Cre}$Tmem173$^{Flox/Flox}$* mice had impaired T$_H$ responses to CDG/PspA immunization

The CDG/PspA immunized *Itgax$^{Cre}$Tmem173$^{Flox/Flox}$* mice also showed dramatically decreased T$_H$1, T$_H$2, and T$_H$17 responses in the ex vivo recall assay on splenocytes (*Figure 9C*). Lungs can form Bronchus associated lymphoid tissue after immunization and initiate an adaptive immune response in situ. We did the recall assay on the lung cells from immunized mice to examine the local immune responses. Similar to the responses in splenocytes, T$_H$1, T$_H$17 and, to a lesser degree, T$_H$2 responses were decreased in lung cells from *Itgax$^{Cre}$Tmem173$^{Flox/Flox}$* mice (*Figure 9D*).

## CDG/PspA immunization did not induce protective immunity in the *Itgax$^{Cre}$Tmem173$^{Flox/Flox}$* mice

Next, we examined the CDG/PspA vaccine induced protective immunity in the *Itgax$^{Cre}$Tmem173$^{Flox/Flox}$* mice. While CDG/PspA immunization significantly lowered the lung bacterial burden in the *Tmem173$^{Flox/Flox}$* mice, it did not alter the bacterial burden from lungs of the *Itgax$^{Cre}$Tmem173$^{Flox/Flox}$*

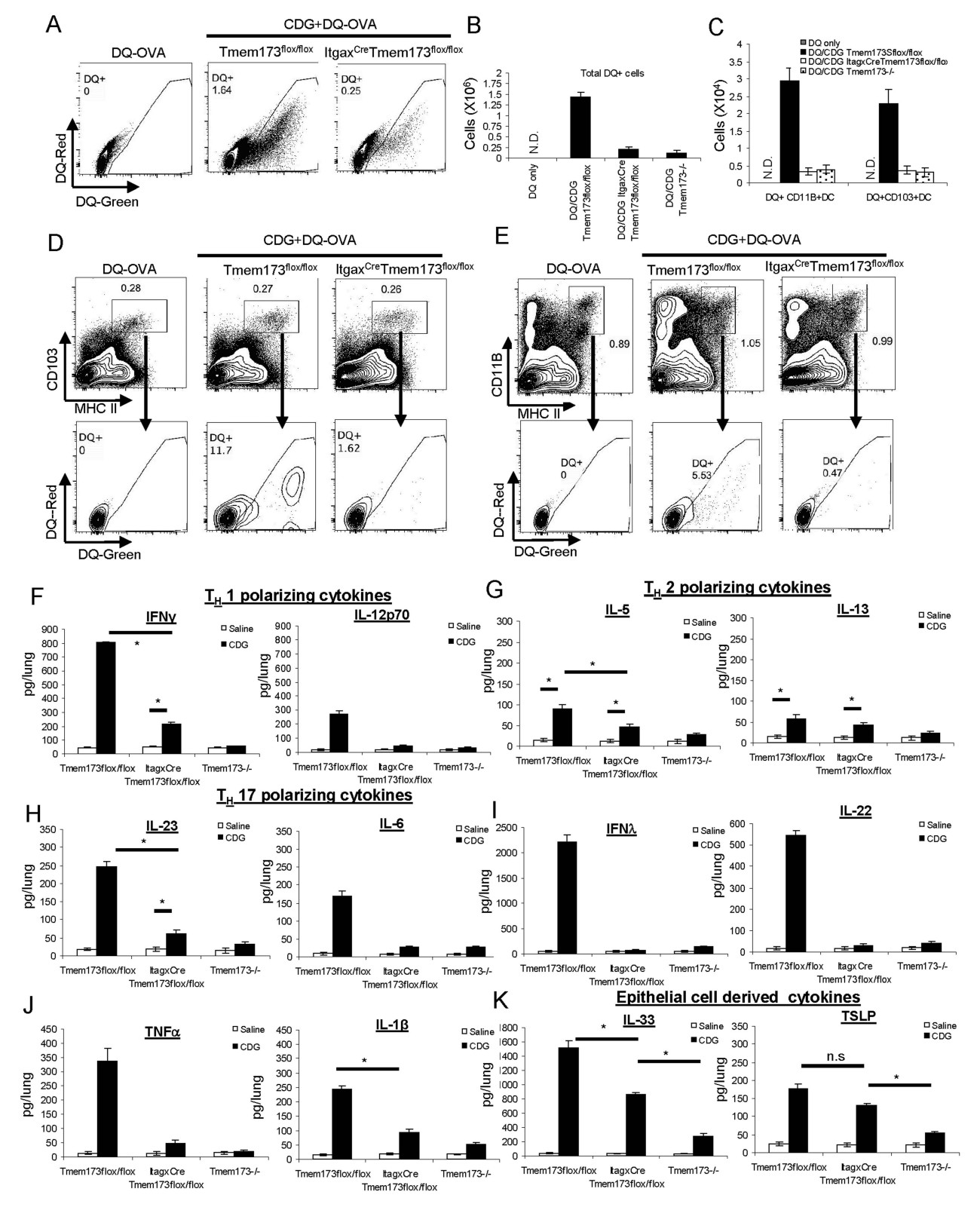

**Figure 8**. CDG induced DC Ag uptake and activation requires MPYS expression in CD11C+ cells. (**A**) Flow cytometry analysis of lung cells from DQ-OVA (20 μg) or DQ-OVA(20 μg) + CDG(5 μg) treated (i.n.) *Tmem173$^{Flox/Flox}$* or *Itgax$^{Cre}$ Tmem173$^{Flox/Flox}$* mice. Live cells were gated. n = 3. (**B**) Total DQ+ lung cells from DQ-OVA or CDG + DQ-OVA treated *Tmem173$^{Flox/Flox}$*, *Itgax$^{Cre}$ Tmem173$^{Flox/Flox}$*, and *Tmem173$^{-/-}$* mice. n = 3. (**C**) DQ+ DCs numbers from DQ-OVA

*Figure 8. continued on next page*

*Figure 8. Continued*

or CDG + DQ-OVA treated *Tmem173^Flox/Flox^*, *Itgax^Cre^ Tmem173^Flox/Flox^* and *Tmem173^−/−^* mice. n = 3. (**D** and **E**) Flow cytometry analysis of DQ⁺ DCs from lung of DQ-OVA or CDG + DQ-OVA treated mice *Tmem173^Flox/Flox^*, *Itgax^Cre^ Tmem173^Flox/Flox^*. Live cells were gated. n = 3. (**F–K**) *Tmem173^Flox/Flox^*, *Itgax^Cre^Tmem173^Flox/Flox^* or *Tmem173^−/−^* mice were treated with saline or CDG (5 µg) for 20 hr. Indicated cytokines were measured in lung homogenates by ELISA. n = 3. Graph present means ± standard error from three independent experiments. Significance is represented by an asterisk, where p < 0.05.

The following figure supplement is available for figure 8:

**Figure supplement 1**. Generation of *Itgax^Cre^Tmem173^Flox/Flox^* mouse.

mice (*Figure 9E*). We concluded that the mucosal pneumococcal vaccine adjuvant activity of CDG requires MPYS expression in CD11C⁺ cells. Noticeably, unlike the *Tmem173^−/−^* mice, PspA immunized *Tmem173^Flox/Flox^* and *Itgax^Cre^Tmem173^Flox/Flox^* mice had similar lung bacterial burden (*Figure 9E*).

## The impaired adjuvant activity of CDG in *Itgax^Cre^Tmem173^Flox/Flox^* mice is not due to the overexpression of *Cre* gene in the CD11C⁺ cells

The *Itgax^Cre^Tmem173^Flox/Flox^* mice also overexpressed the *Cre* gene in the CD11C⁺ cells. To exclude the possibility that the defect seen in the *Itgax^Cre^Tmem173^Flox/Flox^* mice was due to the *Cre* overexpression, we compared *Itgax^Cre^Tmem173^Flox/Flox^* mice with the *Itgax^Cre^*-C57BL/6 mice upon intranasal CDG/PspA immunization. Similar to the observation in *Figure 9*, CDG/PspA immunized *Itgax^Cre^Tmem173^Flox/Flox^* mice had a severe defect in anti-PspA Ab production compared to immunized *Itgax^Cre^*-C57BL/6 mice (*Figure 9—figure supplement 1A*). Their T_H responses in spleen cells were largely non existent, except for IL-5 (*Figure 9—figure supplement 1B*). A similar observation was made in lung recall assay (*Figure 9—figure supplement 1C*).

## Discussion

Our study revealed two novel in vivo mechanisms of action of the mucosal vaccine adjuvant CDG (*Figure 10*). First, CDG enhances Ag uptake in APCs and non-APCs in vivo. Second, CDG activates pinocytosis-efficient cells in vivo. CDG has two phosphate groups preventing it from directly passing through the cell membrane. The mammalian receptor for CDG, MPYS, is located inside cells. Thus, though intranasally administered CDG enhances Ag uptake in all types of cells, only cells that efficiently take up CDG, via pinocytosis, will be activated (*Figure 10A*).

How does CDG, as a mucosal adjuvant, enhance Ag uptake in vivo? Three observations in this study may shed light on the mechanism. First, CDG enhances Ag uptake by both pinocytosis and receptor mediated endocytosis (*Figure 4*); Second, while CDG enhances Ag uptake in all types of cells (*Figure 4*), deletion of MPYS in only CD11C⁺ cells severely impaired that (*Figure 8*); Third, MPYS expression in CD11C⁺ cells is mainly responsible for the CDG induced cytokine milieu in lungs (*Figure 8*). We propose that CDG enhances MPYS-dependent Ag uptake in cells directly taking up CDG (pinocytosis-efficient, OVA-647^Low^, DQ-Green⁺DQ-Red⁻ cells). In cells that do not take up CDG (OVA-647^hi^, DQ-Green⁺DQ-Red⁺ cells), Ag uptake is enhanced by the cytokine milieu generated mainly by CDG activated CD11C⁺ cells (*Figure 10B*).

We favored the hypothesis that intranasally administered CDG directly primed pulmonary DCs, leading to MPYS-dependent production of T_H polarizing cytokines in vivo (*Figure 10C*). Two pieces of data support this hypothesis. First, we detected IL-12 and IFNγ producing DCs in vivo as early as 5 hr post treatment (*Figure 3F,G*). Second, the *Itgax^Cre^Tmem173^Flox/Flox^* mice had dramatically decreased CDG-induced T_H1 and T_H17 cytokine in vivo (*Figure 8F,H*). There are two CD11C⁺ populations in the lung: MHC II^hi^ and MHC II^low^. Among Ag positive (OVA-647⁺) cells, only the MHC II^hi^CD11C⁺ population (i.e., DCs) were activated by CDG in vivo (*Figure 4E*). This suggested that the deletion of MPYS in MHC II^hi^CD11C⁺ cells (DCs) was responsible for the impaired T_H1 and T_H17 polarizing cytokine production in vivo.

Intriguingly, the production of T_H2 polarizing cytokines IL-5 and IL-13 was less dependent on the expression of MPYS in DCs (*Figure 8G*). Indeed, unlike *Tmem173^−/−^* mice, *Itgax^Cre^Tmem173^Flox/Flox^* mice still have some Ab and T_H responses after CDG immunization (*Figure 9*). Therefore besides DCs, MPYS expression in other cells contributes to the adjuvant activity of CDG in vivo. We found that the

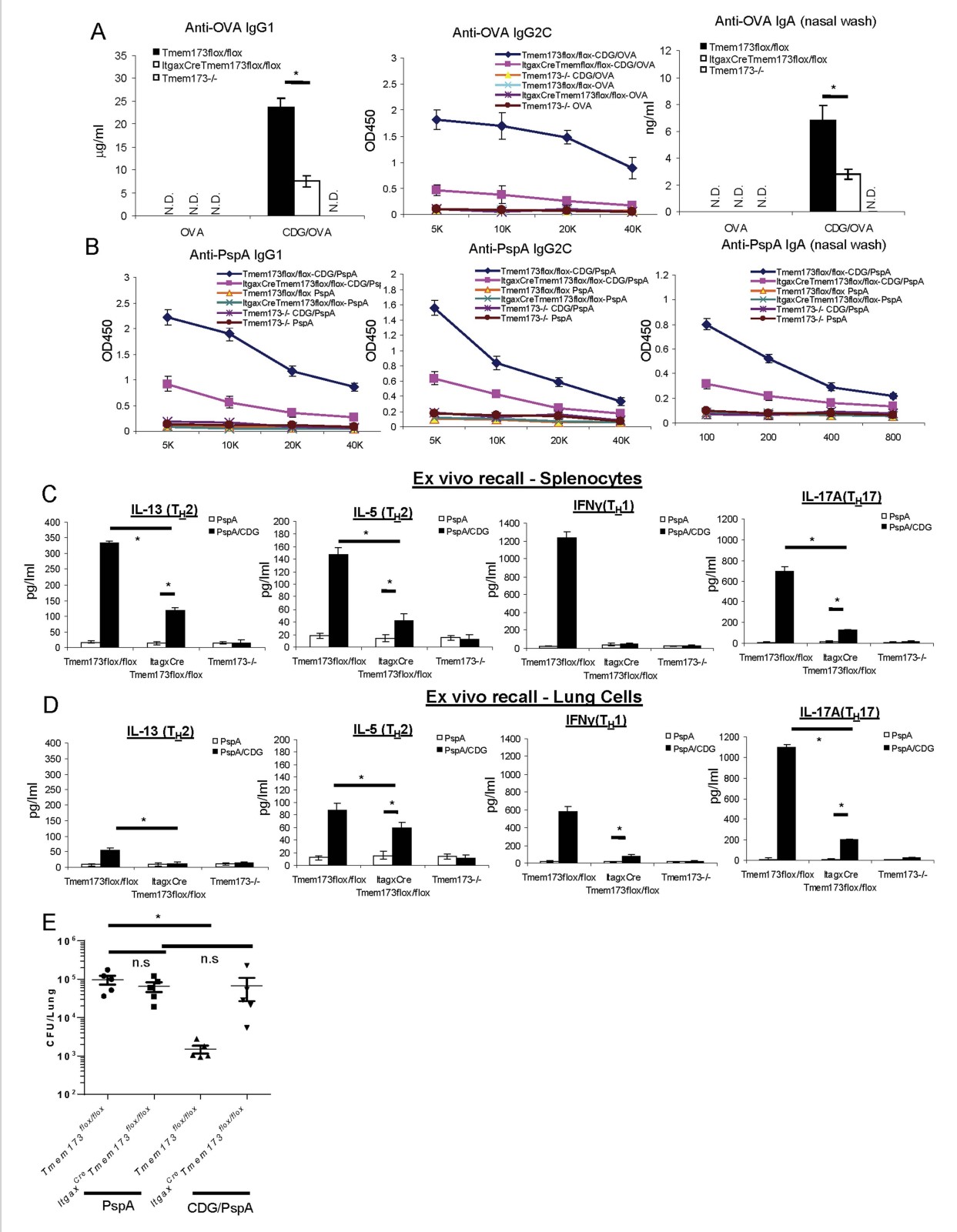

**Figure 9**. MPYS expression in CD11C[+] cells is required for the optimal mucosal adjuvant activity of CDG. (**A**) *Tmem173[Flox/Flox]*, *Itgax[Cre]Tmem173[Flox/Flox]* or *Tmem173[−/−]* mice were intranasally administered OVA (20 μg) alone or together with 5 μg CDG as in *Figure 1A*. Anti-OVA IgG1, IgG2C and IgA were determined by ELISA. n = 3. (**B**) *Tmem173[Flox/Flox]*, *Itgax[Cre]Tmem173[Flox/Flox]* or *Tmem173[−/−]* mice were immunized with PspA (2 μg) alone or together with

*Figure 9. continued on next page*

*Figure 9. Continued*

5 µg CDG as in *Figure 1C*. Anti-PspA IgG1, IgG2C, and IgA were measured by ELISA. n = 3. (**C–D**) Splenocytes and lung cells from PspA or CDG + PspA immunized *Tmem173$^{Flox/Flox}$*, *Itgax$^{Cre}$Tmem173$^{Flox/Flox}$* or *Tmem173$^{-/-}$* mice were stimulated with 5 µg/ml PspA for 4 days in culture. Cytokines were measured in the supernatant by ELISA as in *Figure 1D* n = 3. (**E**) 1 month after the last immunization, CDG/PspA or PspA immunized *Tmem173$^{Flox/Flox}$* and *Itgax$^{Cre}$Tmem173$^{Flox/Flox}$* mice were infected (i.n.) with *S. pneumoniae* (D39 strain, ~5.0 × 10$^6$ c.f.u.). At 48 hr post infection, lung bacterial burden were determined. n = 2. Graph present means ± standard error from three independent experiments. Significance is represented by an asterisk, where p < 0.05.

The following figure supplement is available for figure 9:

**Figure supplement 1**. The impaired CDG response in *Itgax$^{Cre}$Tmem173$^{Flox/Flox}$* mice is not due to the over-expression of Cre in CD11C$^+$ cells.

OVA-647$^+$CD11C$^-$MHC II$^{int}$ APC was activated by CDG in vivo (*Figure 4E*). The total number of OVA-647$^+$ activated cells in this CD11C$^-$MHC II$^{int}$ population are comparable to that of the CD11C$^+$ MHC II$^{hi}$ population (*Figure 4G*). Thus, these CD11C$^-$ OVA-647$^+$ MHC II$^{int}$ CD86$^+$ cells may contribute to the adjuvant activity of CDG in *Itgax$^{Cre}$Tmem173$^{Flox/Flox}$* mice.

How does CDG enhance MPYS-mediated Ag uptake in pinocytosis-efficient cells in vivo? CDG activates MPYS-TBK1-IRF3-Type I IFN signaling in vitro. However, intranasally administered CDG did not induce Type I IFN production in vivo. Instead, it generates type II, type III IFN, and various cytokines that depend on NF-κB activation. We previously showed, in vitro, that CDG induced type I IFN and NF-κB activation can be uncoupled in DCs and macrophages (*Blaauboer et al., 2014*). Thus, MPYS is not just a type I IFN stimulator. New molecular mechanisms by which CDG enhances MPYS-dependent Ag uptakes as well as activation of Type II, III IFN and NF-κB signaling in pinocytosis-efficient cells in vivo remains to be discovered.

*Tmem173$^{-/-}$* mice still made several cytokines, namely IL-1α, IL-1β, IL-6, IL-10, IL-33, and TSLP, after CDG treatment in vivo. This indicates that CDG can activate MPYS-independent signaling in vivo. A previous study showed that CDG activated NLRP3 inflammasome independent of STING/MPYS (*Abdul-Sater et al., 2013*). CDG also bound to hyperpolarization-activated cyclic nucleotide-gated channel 4 (HCN4) and inhibited cAMP regulated heart rate (*Lolicato et al., 2014*). Very recently, it was found that cyclic di-AMP, a similar cyclic dinucleotide for STING/MPYS, induces human monocyte apoptosis independent of STING/MPYS (*Tosolini et al., 2015*). Thus, other mammalian receptors for CDG exist.

We showed that CDG is a superior mucosal pneumococcal vaccine adjuvant than the 2'3'-cGAMP in mice (*Figure 1*). As a mammalian cyclic dinucleotide, 2'3'-cGAMP can be hydrolyzed by the ecto-nucleotide phosphodiesterase (ENPP1) found in mammalian cells (*Li et al., 2014*). On the contrary, as a bacterial cyclic dinucleotide, CDG may be more resistant to hydrolysis when introduced into mammalian cells. Further study is needed to determine if CDG is a better human adjuvant than 2'3'-cGAMP.

The anti-tumor molecules 10-carboxymethyl-9-acridanone (CMA) (*Cavlar et al., 2013*) and 5,6-dimethylxanthenone-4-acetic acid (DMXAA) (*Conlon et al., 2013*) activate mouse, but not, human MPYS signaling. CDG, on the other hand, is functional in human cells as well (*Karaolis et al., 2007*; *Yi et al., 2013*). In fact, CDG binds to mouse MPYS/STING (*Burdette et al., 2011*) and human MPYS/STING (*Ouyang et al., 2012*; *Shang et al., 2012*; *Shu et al., 2012*; *Yin et al., 2012*) with similar dissociation constant ($K_d$: 2–5 µM). Nevertheless, we first discovered that human TMEM173 gene displays great heterogeneity (*Jin et al., 2011b*). We further identified a loss-of-function human TMEM173 variant HAQ (R71H-G230A-R293Q) that is carried by ~20% of Americans (*Jin et al., 2011b*). In vitro studies demonstrated that many of these human TMEM173 variants are functionally different from the R232 (wild type) TMEM173 allele (*Ablasser et al., 2013*; *Gao et al., 2013b*; *Yi et al., 2013*; *Zhang et al., 2013*). To develop CDG or other cyclic dinucleotide as a human mucosal vaccine adjuvant, it becomes critical to determine if the adjuvant activity of cyclic dinucleotides is influenced by human TMEM173 variations in vivo.

In summary, we found that CDG enhances Ag uptake and selectively activates pinocytosis-efficient cells in vivo. These qualities should be explored further for the development of CDG as an effective human mucosal vaccine adjuvant.

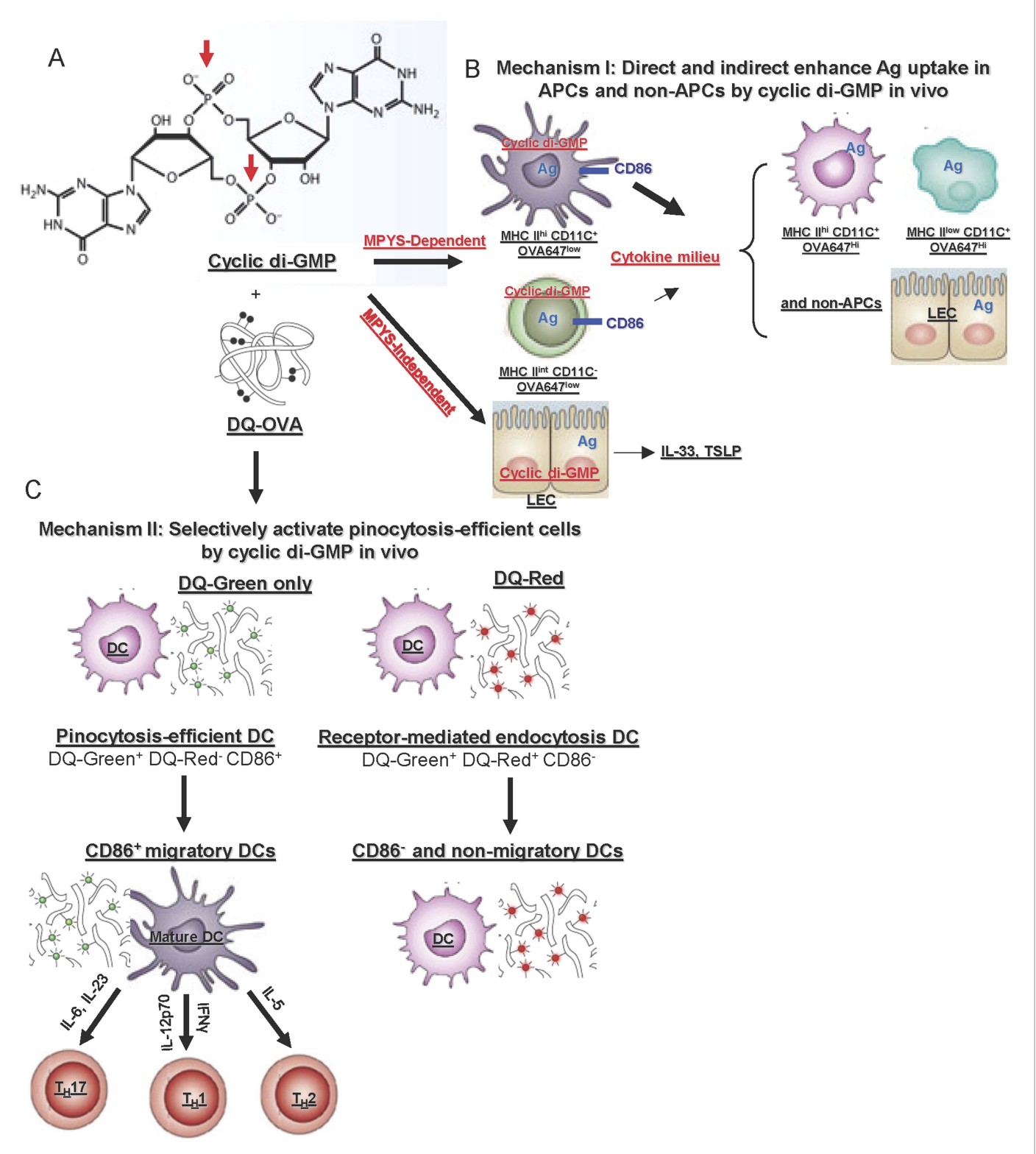

**Figure 10**. In vivo mechanisms of the mucosal vaccine adjuvant CDG. (**A**) The formula of CDG. Red arrows indicate the phosphate groups that prevent CDG from directly crossing the cell membrane. (**B**) Mechanism I: CDG enhances Ag uptakes in APCs and non-APCs. Among OVA647+APCs, only a portion of MHC IIhiCD11C+ (DCs) and MHC IIint CD11C− cells up-regulate CD86 expression in vivo. They are mainly OVA647low cells, which take up Ag by pinocytosis. The activation of these cells generate a cytokine milieu that acts on other cells leading to enhanced Ag uptake (OVA647hi cells) but not cell

*Figure 10. continued on next page*

*Figure 10. Continued*

activation. CDG also activates lung epithelial cells (LEC), leading to TSLP and IL-33 production. But this is only partially dependent on MPYS and is not sufficient to enhance Ag uptake in vivo (*Figure 8*). (**C**) Mechanism II: CDG selectively activates pinocytosis-efficient DCs in vivo. After administering DQ-OVA together with CDG, the DQ+MHC IIhiCD11C+ Ag-loading DCs can be separated into two distinct populations: DQ-Green+DQ-Red−CD86+ and DQ-Green+DQ-Red+CD86−. DQ-Green+DQ-Red− cells are OVA-647low cells, while DQ-Green+DQ-Red+ cells are OVA-647hi cells. Only the DQ-Green+ DQ-Red−CD86+ cells migrated to lung DLNs.

## Materials and methods

### Mice

6–12 week old mice were used for all experiments. *Tmem173−/−* mice (Tmem173<tm1Camb>) have been described previously (*Jin et al., 2011a*, *2013*). The *Itgax^Cre^Tmem173^Flox/Flox^* mouse was generated as in *Figure 4A*. All mice are on a C57BL/6 background. Mice were housed and bred in the Animal Research Facility at Albany Medical College. All experiments with mice were performed in accordance to the regulations and approval of Albany Medical College (Albany, NY) and the Institutional Animal Care and Use Committee.

### Reagent

The following reagent was obtained through BEI Resources, NIAID, NIH, Bethesda, MD:*S. pneumoniae* Family 1, Clade 2 PspA (UAB055) with C-Terminal Histidine Tag, Recombinant from *Escherichia coli*, NR-33178.

### Intranasal immunization

Mice were immunized with three doses (14 days apart) of OVA (20 µg, cat# vac-efova; Invivogen, San Diego, CA) or PspA (2 µg, BEI Resources) with, or without, CDG (5 µg, cat# vac-cdg; Invivogen) or 2′3′-cGAMP (5 µg, cat# vac-cga23; Invivogen). Groups of mice (4 per group) were intranasally vaccinated with adjuvanted protein Ag, or Ag alone. For intranasal vaccination, animals were anaesthetized using isoflurane in an E-Z Anesthesia system (Euthanex Corp, Palmer, PA). Ag, with or without CDG, was administered. Sera and nasal washes were collected 14 days after the last immunization.

### Detection of Ag-specific Ab

The Ag-specific Abs were determined by ELISA. The anti-IgG-HRP used was anti-mouse IgG1-HRP (cat#1070-05; Southern Biotech, Birmingham, AL), anti-mouse IgG2C-HRP (cat#1079-05; Southern Biotech), and anti-mouse IgA-HRP (cat#1040-05; Southern Biotech). Total anti-OVA IgG2A, IgG1 and IgA were quantified using a mouse anti-OVA IgG2A kit (cat#3015; Chondrex, Redmond, WA), anti-OVA IgG1 kit (cat#3013; Chondrex, Redmond, WA), and anti-OVA IgA kit (cat#3018; Chondrex, Redmond, WA).

### Bronchoalveolar lavage

Mice were sacrificed at the indicated time by $CO_2$ asphyxiation and lungs were lavaged with 0.8 ml ice-cold PBS. The lavage fluid was centrifuged at 2000×*g* for 1 min. Collected cells were analyzed by Flow cytometry.

### Detection of lung cytokine production

Mice were intranasally administered 5 µg CDG (vaccine grade), then sacrificed at the indicated time by $CO_2$ asphyxiation. BALF was collected and the lungs were subsequently perfused with cold PBS. The harvested lungs were washed in PBS once, then stored in 0.7 ml tissue protein extraction reagent (T-PER) (cat#78510; Thermo Scientific, Grand Island, NY) containing protease inhibitors (cat#11836153001; Roche, Indianapolis, IN) at −80°C. Later, the lung was thawed on ice and homogenized on ice in the T-PER homogenate buffer in a 2 ml homogenizer. Lung homogenates were transferred to a 1.5 ml tube and spun at 14,000×*g* for 30 min at 4°C. Supernatant was collected and analyzed for cytokine production.

### *Streptococcus pneumoniae* infection

*S. pneumoniae* D39 (serotype 2; ATCC, Manassas, VA) were grown in Todd-Hewitt broth containing 0.5% yeast extract (THY; BD Biosciences, San Jose, CA) to an optical density (OD) of 0.4 (~$10^8$ cfu/ml). Mice were intranasally administered ~$5 \times 10^6$ cfu. CFUs were confirmed by colony counting of $\log_{10}$

serial dilutions of bacteria cultured overnight on a TSA II with 10% sheep blood agar plate (cat#221162; BD Bioscience).

## Flow cytometry analysis of in vivo Ag uptake and processing

Mice were intranasally administered 20 µg DQ-Ovalbumin (DQ-OVA) (D12053; Life Technologies, Grand Island, NY) or 20 µg Ovalbumin Alexa Fluor 647 (OVA-647) (O34784; Life Technologies) with, or without, the adjuvant CDG (5 µg, vaccine-grade). After 20 hr, the lungs were lavaged and perfused with ice-cold PBS. Excised lungs were digested in DMEM contain 200 µg/ml DNase I (10104159001; Roche), 25 µg/ml Liberase TM (05401119001; Roche) at 37°C for 3 hr. Red blood cells were then lysed and a single cell suspension was prepared and analyzed by BD LSR II and FACScan flow cytometry.

The following Abs from Biolegend, San Diego, CA were used in the flow cytometry: CD80 (16-10A1), CD86 (GL1), Ly6C (HK1.4), CD11B (M1/70), Ly6G (1A8), IgD (11-26c.2a), CD11C (N418), FcεRIa (MAR-1), NK-1.1 (PK136), MHC II (M5/114.15.2), CD103(2E7). The following Abs from BD Biosciences were used: Siglec F (E50-2440), c-Kit (2B8), and CD68 (FA-11).

## Intracellular IL-12p35 and IFNγ staining

The intracellular cytokine staining was performed using the Cytofix/Cytoperm kit from BD Biosciences (cat#555028). Briefly, mice were intranasally administered saline or CDG (5 µg, vaccine-grade). The lungs were lavaged, perfused, and harvested at 5 hr post treatment. Excised lungs were washed in PBS and digested in DMEM containing 200 µg/ml DNase I (10104159001; Roche), 25 µg/ml Liberase TM (05401119001; Roche), and Golgi-plug at 37°C for 6 hr. The single lung cell suspension was fixed in Cytofix/perm buffer (BD Biosciences) in the dark for 20 min at RT. Fixed cells were then washed and kept in Perm/wash buffer at 4°C. Golgi-plug was present during every step before fixation. The following Abs from eBioscience were used: IL-12p35 (4D10P35) and IFNγ (XMG1.2).

## Cytokine ELISAs

Cytokine concentrations were measured using ELISA kits from eBioscience, San Diego, CA according to the manufacturer's instructions. The ELISA kits used were IL-1α (cat#88-5019), IL-1β (cat#88-7013), IL-4 (cat#88-7044), IL-5 (cat#88-7054), IL-6 (cat#88-7064), IL-10 (cat#88-7105), IL-12/p70 (cat#88-7921), IL-13 (cat#88-7137), IL-17A (cat#88-7371), IL-22 (cat#88-7422), IL-23 (cat#88-7230), IL-33 (cat#88-7333), TNF-α (cat#88-7324), TGF-β1 (cat#88-8350), TSLP (cat#88-7490), IFN-λ (cat#88-7284), and IFN-γ (cat#88-7314). The IFNβ ELISA kit was from PBI InterferonSource, Piscataway, NJ (cat#42410-1).

## Statistical analysis

All data are expressed as means ±SEM. Statistical significance was evaluated using Prism 5.0 software to perform a Student's $t$-test (unpaired, two tailed) for comparison between mean values.

## Online supplemental figures

The online supplemental materials include 3 supplemental figures.

## Acknowledgements

We thank the Flow Cytometry Core Facility in the Center for Immunology and Microbial Diseases for the assistance.

## Additional information

### Funding

| Funder | Grant reference | Author |
|---|---|---|
| National Institute of Allergy and Infectious Diseases (NIAID) | 1R56AI110606-01 | Lei Jin |
| National Institute of Allergy and Infectious Diseases (NIAID) | 1R01AI110606-01A1 | Lei Jin |
| Albany Medical College | New Faculty Start-up Fund | Lei Jin |

The funders had no role in study design, data collection and interpretation, or the decision to submit the work for publication.

## Author contributions
SMB, SM, HRT, HLW, VDG, Acquisition of data, Analysis and interpretation of data; LJ, Conception and design, Acquisition of data, Analysis and interpretation of data, Drafting or revising the article

## Ethics
Animal experimentation: All experiments with mice were performed in accordance to the regulations and approval of Albany Medical College (Albany, NY) and the Institutional Animal Care and Use Committee, ACUP NO: 1208002.

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
