## [Decision Letter]

Thank you for sending your work entitled “The Mucosal Adjuvant Cyclic di-GMP Enhances Antigen Uptake and Selectively Activates Pinocytosis-efficient Cells in vivo” for consideration at *eLife*. Your article has been favorably evaluated by Tadatsugu Taniguchi (Senior editor), a Reviewing editor, and two reviewers.

The Reviewing editor and the reviewers discussed their comments before we reached this decision, and the Reviewing editor has assembled the following comments to help you prepare a revised submission.

This is a very comprehensive and thorough study of mucosal responses to cyclic-di-GMP, a potent vaccine adjuvant. The ability of cyclic-di-GMP and related cyclic-di nucleotides to bind and activate STING, an ER membrane protein involved in type I IFN responses is well appreciated. Most of the work to date however in this area has focused on the mechanism of action of cyclic-di GMP and STING signaling in cells in culture. This study explores the in vivo mucosal adjuvant activity of cyclic-di GMP and the underlying mechanisms involved. In an elegant series of experiments the authors compare cyclic-di-GMP with cyclic GMP-AMP (cGAMP) an endogenous second messenger also known to bind STING. The authors provide evidence for the superior adjuvant activity of the bacterial ci-di-nucleotide over cGAMP. Moreoever, they provide evidence that cyclic di-GMP promotes its adjuvant activity in vivo by enhancing Ag uptake and via its ability to mobilize mucosal DCs in vivo, to promote its adjuvant effects. The authors generated CD11c-specific deletion of STING and uncover DC intrinsic roles for STING in mediating the adjuvant activities observed. Several really interesting aspects come out of these studies. Firstly, the authors convincingly demonstrate that the effects of cyclic-di-GMP are independent of type I IFN, but rather involve type II and type III IFNs. Secondly, their studies indicate that some in vivo effects of cyclic-di-GMP proceed independently of STING. These latter observations indicate that there are likely additional cyclic-di-GMP receptors important in vivo.

Specific comments:

1) STING-dependency is shown only for immune correlates (for example; cytokines, antigen uptake, etc). Whether actual immunity to a pathogen (after immunization with CDN adjuvants) requires STING has not been tested in this study. It would therefore be interesting to challenge STING-deficient mice (both germline knockout and conditional CD11c-Cre) with *S.pneumoniae* following PspA+c-di-GMP vaccination (as done in Figure 1 with WT mice). The STING is indeed crucial to successful vaccination with CDNs, pathogen burden in STING knockouts should be similar to that seen in WT mice vaccinated with non-adjuvanted PspA. This experiment is important, as it would implicate STING in the actual outcome/effectiveness of CDN vaccination rather than in immune correlates of the immunization.

2) The observation that the responsiveness of STING for CDNs varies between species raises some concern. C-di-GMP (the CDN used in this study) is far more potent in mouse cells in vitro compared to human cells (for example, Cavlar et al., EMBO J 2013, PMID 23604073, Figure 4). Therefore, whether c-di-GMP is superior to cGAMP as an adjuvant in humans is questionable. It is important that the authors discuss this point and amend the Abstract and other parts of the text to make it clear that c-di-GMP elicits stronger responses than cGAMP *in mice*. It is also noteworthy that different STING alleles found in humans encode proteins responding differentially to cGAMP isoforms (for example, Gao et al., Cell 2013, PMID 23910378).

Minor comments:

1) Figure 4: To help those readers who are unfamiliar with the DQ-OVA^+^OVA647 assay, could the authors include a cartoon of how these compounds work here?

2) Figure 8: Have the authors measured epithelial cell derived cytokines (as in Figure 3)? These should not be affected by knockout of STING in CD11c^+^ cells.

---

## [Author Response]

*1) STING-dependency is shown only for immune correlates (for example; cytokines, antigen uptake, etc). Whether actual immunity to a pathogen (after immunization with CDN adjuvants) requires STING has not been tested in this study. It would therefore be interesting to challenge STING-deficient mice (both germline knockout and conditional CD11c-Cre) with* S.pneumoniae *following PspA+c-di-GMP vaccination (as done in*
Figure 1
*with WT mice). The STING is indeed crucial to successful vaccination with CDNs, pathogen burden in STING knockouts should be similar to that seen in WT mice vaccinated with non-adjuvanted PspA. This experiment is important, as it would implicate STING in the actual outcome/effectiveness of CDN vaccination rather than in immune correlates of the immunization*.

As requested, we now included the pneumococcal challenge studies done on CDG/PspA or PspA immunized WT vs *Tmem173*^*-/-*^ mice (Figure 7) and WT vs *Itgax*^*Cre*^*Tmem173*^*Flox/Flox*^ mice (Figure 9). As expected, there is no significantly CFU difference between CDG/PspA and PspA immunized *Tmem173*^*-/-*^ mice. However, we found that PspA immunized *Tmem173*^*-/-*^ mice have significantly lower CFU than PspA immunize WT mice (Figure 7). We further found that *Tmem173*^*-/-*^ mice, without PspA immunization, are more resistant to *S. pneumoniae* infection than the WT mice (see Figure 11). This is a novel unexpected finding and points to a detrimental role of MPYS in host defense against *S. pneumoniae* infection. Currently, we are actively pursuing this novel and exciting discovery.

Author response image 1.**DOI:**
http://dx.doi.org/10.7554/eLife.06670.018

Similarly, CDG/PspA immunization did not lower bacterial burden in the *Itgax*^*Cre*^*Tmem173*^*Flox/Flox*^ mice (Figure 9). However, differently from the *Tmem173*^*-/-*^ mice, PspA immunized *Tmem173*^*Flox/Flox*^ mice had similar bacterial burden as the PspA immunized *Itgax*^*Cre*^*Tmem173*^*Flox/Flox*^ mice (Figure 9). We further found that *Itgax*^*Cre*^*Tmem173*^*Flox/Flox*^ mice, without PspA immunization, were not resistant to *S. pneumoniae* infection (see Figure 12). This suggested that MPYS expression in DCs and alveolar macrophages are not responsible for its detrimental role in host response to *S. pneumoniae* infection. Currently, we are actively pursuing this novel and exciting discovery.

Author response image 2.**DOI:**
http://dx.doi.org/10.7554/eLife.06670.019

*2) The observation that the responsiveness of STING for CDNs varies between species raises some concern. C-di-GMP (the CDN used in this study) is far more potent in mouse cells in vitro compared to human cells (for example, Cavlar et al., EMBO J 2013, PMID 23604073,*
Figure 4*). Therefore, whether c-di-GMP is superior to cGAMP as an adjuvant in humans is questionable. It is important that the authors discuss this point and amend the Abstract and other parts of the text to make it clear that c-di-GMP elicits stronger responses than cGAMP* in mice*. It is also noteworthy that different STING alleles found in humans encode proteins responding differentially to cGAMP isoforms (for example, Gao et al., Cell 2013, PMID 23910378)*.

We have amended the manuscript to emphasize that our observation was made in mice. We further added two paragraphs in the Discussion to address the issues raised here.

*Minor comments*:

*1)*
Figure 4*: To help those readers who are unfamiliar with the DQ-OVA*^*+*^*OVA647 assay, could the authors include a cartoon of how these compounds work here?*

Yes. We added Figure 4 to illustrate the mechanism of action of DQ-OVA and OVA-647. Thanks for the suggestion.

*2)*
Figure 8*: Have the authors measured epithelial cell derived cytokines (as in*
Figure 3*)? These should not be affected by knockout of STING in CD11C*^*+*^
*cells*.

We added Figure 8 showing the IL-33 and TSLP production in lung from cyclic di-GMP treated *Itgax*^*Cre*^*Tmem173*^*Flox/Flox*^ mice. The lung TSLP production was slightly lower in cyclic di-GMP treated CD11C^Cre^MPYS^flox/flox^ mice than in the *Tmem173*^*Flox/Flox*^, but it was not statistically significant (Figure 8). However, the lung IL-33 production was significantly lower in cyclic di-GMP treated *Itgax*^*Cre*^*Tmem173*^*Flox/Flox*^ mice than in the *Tmem173*^*Flox/Flox*^ (Figure 8).

MPYS expression in CD11C^+^ cells is responsible for the proinflammatory cytokines TNFα and IL-1β production, which can act on lung epithelial cells (Figure 8). Furthermore IFN-λ receptors are uniquely expressed on lung epithelial cells and *Itgax*^*Cre*^*Tmem173*^*Flox/Flox*^ mice lack cyclic di-GMP induced IFN-λ production (Figure 8). We favored the idea that there is a crosstalk/communication between lung epithelial cells and CD11C^+^ cells during cyclic di-GMP induced immune response.